

# High-resolution physicochemical dataset of atmospheric aerosols over the Tibetan Plateau and its surroundings

**Jianzhong Xu[1,2*], Xinghua Zhang[2,a*], Wenhui Zhao[2,3,b], Lixiang Zhai[2,3,c], Miao Zhong[2,3], Jinsen Shi[4], Junying Sun[5], Yanmei Liu[2,d], Conghui Xie[2,e], Yulong Tan[2,3], Kemei Li[2,3], Xinlei Ge[6], Qi Zhang[7], Shichang Kang[2,3,8]**

[1]School of Oceanography, Shanghai Jiao Tong University, Shanghai 200030, China.

[2]State Key Laboratory of Cryospheric Sciences, Northwest Institute of Eco-Environment and Resources, Chinese Academy of Sciences, Lanzhou 730000, China

[3]University of Chinese Academy of Sciences, Beijing 100049, China

[4]Key Laboratory for Semi-Arid Climate Change of the Ministry of Education, College of Atmospheric Sciences, Lanzhou University, Lanzhou 730000, China

[5]Key Laboratory of Atmospheric Chemistry of CMA, Chinese Academy of Meteorological Sciences, Beijing 100081, China

[6]Jiangsu Key Laboratory of Atmospheric Environment Monitoring and Pollution Control, Collaborative Innovation Center of Atmospheric Environment and Equipment Technology, School of Environmental Science and Engineering, Nanjing University of Information Science and Technology, Nanjing 210044, China

[7]Department of Environmental Toxicology, University of California, Davis, CA 95616, USA

[8]CAS Center for Excellence in Tibetan Plateau Earth Sciences, Beijing 100085, China

[a]now at: College of Urban and Environmental Sciences, Northwest University, Xi'an 710127, China.

[b]now at: Institute of Geochemistry, Chinese Academy of Sciences, Guangzhou 510640, China.

[c]now at: College of Resources and Environmental Sciences, Gansu Agricultural University, Lanzhou 730000, China.

[d]now at: College of Resources and Environment, Aba Teachers University, Wenchuan 623002, China.

[e]now at: State Key Joint Laboratory of Environmental Simulation and Pollution Control, College of Environmental Sciences and Engineering, Peking University, Beijing 100871, China.

*Correspondence to*: Jianzhong Xu (jzxu78@sjtu.edu.cn; jzxu@lzb.ac.cn) and Xinghua Zhang (zhangxinghua@lzb.ac.cn)





## Abstract

Atmospheric aerosol in the Tibetan Plateau (TP) and its surroundings has received widely scientific concern in recent decades owing to its significant impacts on regional climatic and cryospheric changes, ecological and environmental securities, and hydrological cycle. However, our understanding on the atmospheric aerosol in this remote region is highly limited by the scarcely available dataset due to the extremely harsh natural conditions. This condition has been improved in recent decades by constructing a few stable field observatories at typical sites on the TP and its surroundings. A continuous project was carried out since 2015 to investigate the properties and sources of atmospheric aerosols as well as their regional differences in the vast TP regions by performing multiple short-term intensive field observations using a suite of high-resolution online instruments. This paper presents a systematic dataset of the high-time-resolution (hourly scales) aerosol physicochemical and optical properties at eight different sites over the TP and its surroundings from the observation project, including the size-resolved chemical compositions of submicron aerosols, standard high-resolution mass spectra and sources of organic aerosols, size distributions of particle number concentrations, particle light scattering and absorption coefficients, particle light absorptions from different carbonaceous substances of black carbon and brown carbon, and number concentrations of cloud condensation nuclei. In brief, atmospheric aerosols in these remote sites were all well-mixed and highly aged due to their dominated regional transport sources. However, high contributions of carbonaceous organic aerosols, neutralized bulk submicron aerosols, and relatively higher light absorption capability were observed in the southern TP region, whereas secondary inorganic species contributed dominantly to the overall acidic submicron aerosols in the northern TP region. In addition to the insights into the regional differences on aerosol sources and properties in the vast TP regions, the datasets are also useful for the simulation of aerosol radiative forcing and the evaluation of interactions among different components of the Earth system in numerical models. The datasets are available from the National Cryosphere Desert Data Center, Chinese Academy of Sciences (https://doi.org/10.12072/ncdc.NIEER.db2200.2022; Xu, 2022).



## 1 Introduction

Tibetan Plateau (TP), with a mean altitude of over 4000 m a.s.l. and a huge surface area of approximately $2.5 \times 10^6$ km$^2$, is the highest plateau on the Earth. The high-altitude mountain ranges on the TP and its surroundings are one of the most important cryospheric regions in the world. Therefore, the TP has been widely known as the "roof of the world", the "Third Pole", or the "Asian Water Tower" (Qiu, 2008; Yao et al., 2019). The TP and its surroundings have significant impacts in the global and regional climate systems, hydrological cycles, and cryospheric changes through its huge and complex topography and heat source (Duan and Wu, 2005; Yao et al., 2012; Chen et al., 2021). Over the past few decades, one of the most concerns on the TP and its surroundings has been focusing on the significant climatic warming and rapid cryospheric changes in this region (Kang et al., 2010), which show a higher warming rate than the Northern Hemisphere (0.34 vs. 0.29 °C/decade) (You et al., 2021; Zhou and Zhang, 2021).

As the most complex and important component in the atmosphere, atmospheric aerosols play significant roles in the climatic warming and cryospheric changes in the TP regions through their crucial direct and indirect effects on solar radiation and the albedos of snow and ice surfaces (Xu et al., 2009; Kang et al., 2019b). Atmospheric aerosols, particularly the two important light-absorbing carbonaceous aerosols (CAs) of black carbon (BC) and brown carbon (BrC), can absorb the solar radiation directly, warm the atmosphere, and finally lead to a positive forcing on Earth's energy budget (Ramanathan et al., 2007; Kopacz et al., 2011). For example, Li et al. (2018) has simulated significantly higher average albedo reduction (~46%) and instantaneous radiative forcing (7–64 W m$^{-2}$) caused by BC than mineral dust that deposited on aged snow in the surface of a TP glacier. In addition, aerosol particles over the TP also exert significant impacts on ice cloud properties and cloud development through their semi-direct effects (Liu et al., 2019). Since the direct observation of atmospheric aerosols over the vast and remote TP regions is remarkably difficult due to the complex topography and meteorology and harsh environment, the numerical model simulation based on reanalysis data has become one of the most popular and critical ways over the past decades. For example, Lau et al. (2006) evaluated the significant impact of atmospheric aerosols over the TP on the intensification of the Asian summer monsoon by using the NASA finite-volume general circulation model, of which inputs were



driven by the realistic global wind analyses and humidity; Kopacz et al. (2011)
investigated the origin and radiative forcing of BC transported to the TP and Himalayas
by using the GEOS-Chem global chemical transport model based on the global BC
emissions inventory data; Liu et al. (2015) investigated the transport of summer dust
and anthropogenic aerosols over the TP by using a three-dimensional aerosol transport–
radiation model with the inputs from diverse satellite remoting products. Although
important findings have been reported from those numerical model simulations, the in-
situ observation of atmospheric aerosols over the TP regions has become more crucial
and urgent due to their key roles in evaluating and improving the model performances
over the remote region. Lacking in-situ observational aerosol parameters for
constraining the model would lead to high uncertainty in the results. In addition, model
simulations in the TP mostly focused on the spatial distribution over a large range, but
rarely on the temporal variations or inherent evolution mechanism with high time
resolution as those from in-situ observations.
With great improvements in observational conditions and instruments, numerous in-
situ measurements have been conducted in the TP and its surroundings during the past
few years to characterize the aerosol physical, chemical, and optical properties,
potential sources, transport pathways, and regional distributions. A comprehensive
summary of direct measurements on ambient aerosols in the TP using various
observational methods and instruments has been listed in Table S1 in the supplementary.
Among them, the off-line atmospheric filter sampling was one of the most important
and popular in-situ aerosol collection approach in the TP because it was relatively low-
cost and easy to carry out under the extremely harsh natural conditions and limit
logistical supports. This approach has been successfully conducted to characterize the
compositions, sizes, light absorptions, sources, and variations of ambient aerosols,
especially those different CAs constituents including BC, BrC, organic carbon (OC),
water-soluble OC (WSOC), humic-like substances (HULS), and polycyclic aromatic
hydrocarbons (PAHs), in the remote TP region (Cao et al., 2009; Zhao et al., 2013; Xu
et al., 2014a; Zhang et al., 2014; Cong et al., 2015; Wan et al., 2015; Xu et al., 2015;
Kang et al., 2016; Li et al., 2016b; Xu et al., 2020). In addition, characterizing the
regional distribution of atmospheric aerosols over a large area of the TP was another
important advantage of the off-line filter sampling through their simultaneous
observations at multiple sites due to the relatively easy instrumental operations (Li et





al., 2016a; Chen et al., 2019; Kang et al., 2022). However, studies on the atmospheric
aerosols over the remote TP regions through these off-line filter samplings are still
scarce and far meeting the demand at present. These observational data was generally
scattered and unsystematic with relatively low time resolution, few aerosol property
parameters, and less data points because most of them were carried out at certain site
using single instruments with time resolutions from days to weeks during a short-term
period. The low time resolution measurement would limit the accurate understanding
of temporal evolution processes and mechanisms of aerosol properties during a fast and
short-term event. Meantime, these observational data from different research groups
was also difficult to compare and integrate because most of them had different research
concerns, measurement parameters, sampling flows, laboratory filter treatment
procedures, data analysis methods, etc. Besides, although the off-line sampling is
relatively easy to carry out, there are still large TP regions where similar observations
have not yet been conducted or only conducted in a short-term period. Up to now,
comprehensive research focusing on multiple aerosol physicochemical and optical
properties through the real-time online consecutive measurements (high temporal
resolutions from minute to hour scales) at multiple sites are still rare in the TP.
Studies on atmospheric aerosols have achieved great progress during the last decade in
China. The Aerodyne high-resolution time-of-flight aerosol mass spectrometer (HR-
ToF-AMS) was one of the most popular online instrument that successfully
implemented in numerous aerosol chemistry observations to characterize the real-time
size-resolved chemical compositions and sources of submicron aerosols (Li et al., 2017;
Zhou et al., 2020). However, the utilization of HR-ToF-AMS on the TP was still very
scarce due to the high instrumental requirements but extremely harsh observation
conditions. Since 2015, a continuous and systematic observation project, aiming to
investigate the regional differences on aerosol sources and properties in the different
TP regions, has been launched by our research team by performing HR-ToF-AMS
measurement and other high-resolution real-time online instruments at different sites
during almost every year. Indeed, our dataset was the first and only set of dataset
focusing on abundant aerosol parameters (including the physical, chemical, and optical
properties and diverse sources) at multiple different types of site (e.g., the urban, remote,
high-altitude mountain, grassland, and subtropical forest sites that basically represented
the different TP regions) based on the real-time high-time-resolution online





observations in the TP and its surroundings. These datasets provide not only
comprehensive data for the understanding of regional differences in aerosol sources and
properties in different TP regions, but also potential basic inputs for the simulation of
aerosol radiative forcing and the assessments of interactions among different
components of the Earth system in future models. The contents of this paper are
organized as: Sects. 2 and 3 describe the observation sites, instrumental deployments,
and data processing methods, Sect. 4 introduces the high-time-resolution data of aerosol
physical, chemical, and optical properties as well as sources, while the limitations and
uniqueness of our dataset are discussed in Sect. 5.

## 2 Observation site descriptions

Continuous observations of atmospheric aerosol chemistry were conducted at eight
sites in the different regions of TP and its surroundings from 2015 to 2022. The sites
include seven remote sites (QOMS, Motuo, NamCo, Ngari, Waliguan, LHG, and
Bayanbulak) and one urban site (Lhasa) which is used for comparison. Figure 1
illustrates the geographical locations of these sites and the picture of each observation.
Table 1 provides detailed information of each site as well as the sampling period and
available instruments during each field campaign. The following text gives a brief
description of each site from the south to north of the TP.

### 2.1 QOMS

The Qomolangma Station for Atmospheric and Environmental Observation and
Research, Chinese Academy of Sciences (QOMS for short in this study and similarly
hereinafter for other sites; 86.56ºE, 28.21ºN; 4276 m a.s.l.) is situated in the basin of
Rongbuk valley in the northern slope of Mt. Everest. The climate in the northern slope
of Mt. Everest has obvious seasonal variation under the influence of Indian monsoon
system (Bonasoni et al., 2010; Cong et al., 2015). During the pre-monsoon season
(generally March−May), the dominated westerlies play important roles in the long-
range transport of atmospheric pollutants from those polluted regions in South Asia,
which makes QOMS an ideal high-altitude observatory at the south edge of the TP for
studying the transboundary transport of atmospheric pollutants from South Asia to the
inner TP. During the monsoon season in summer (June−August), the southerly winds
prevail and bring warm and wet airflow from the Indian Ocean to this region with
increasing humidity and precipitation.



### 2.2 Motuo

Motuo county is in the lower reaches of the Yarlung Tsangpo River and the southern slope of the eastern Himalayas and Gangrigab Mountains in the southeast edge of the TP. The county sits at a halfway up a mountain and has a subtropical humid climate with relatively high temperature and abundant rainfall. Due to the minor population (~15,000), Motuo county is also a relatively pristine region in the TP. The sampling site in Motuo (29.30°N, 95.32°E; 1305 m a.s.l.) was at the summit of a hill that towards to the Yarlung Tsangpo Grand Canyon, which makes it a very ideal site in the southeast edge of the TP to directly monitor the transboundary transport of atmospheric pollutants and moisture from Southeast Asia and Indian Ocean to the TP.

### 2.3 NamCo

The Nam Co Station for Multisphere Observation and Research, Chinese Academy of Sciences (NamCo; 90.95ºE, 30.77ºN; 4730 m a.s.l.) is a high-altitude observatory at the south-central part of the TP. This station is situated at the southeast shore of Nam Co Lake. The surrounding of this station is a pristine region in the TP and isolated from major populated areas. The region is generally affected by the typical semi-arid plateau monsoon climate with more precipitation during the summer monsoon season. The NamCo is the most important site in inland of the TP and dominated by air mass from south and west.

### 2.4 Ngari

The Ngari Station for Desert Environment Observation and Research, Chinese Academy of Science (Ngari; 79.70°E, 33.39°N; 4270 m a.s.l.) locates in the Rutog County, Ngari Prefecture, Tibet Autonomous Region, China. This region is the southwestern edge of the TP and belongs to semi-arid areas with barren land surface and strong solar radiation. As the major member of the "high-cold region observation and research network for land surface processes & environment of China", Ngari station is one of the most important filed stations for monitoring the changes in climate, hydrology, atmosphere, and ecological environment in the western region of the TP and further revealing the interaction between the Indian monsoon and the westerly belt.

### 2.5 Waliguan

The Waliguan Baseline Observatory (Waliguan; 100.9ºE, 36.28ºN; 3816 m a.s.l.) is one of the twenty-nine baseline stations of Global Atmosphere Watch (GAW) of the World





224 Meteorological Organization (WMO). The observatory is situated at the top of the Mt.

225 Waliguan (mountain height of ~600 m), which is also a relatively pristine region with

226 little influence from human activities. The Waliguan is an important observatory in the

227 northeast edge of the TP and dominated by air mass from northeast during the summer

228 season, which makes it an ideal site to study the influence of air pollutants from the

229 industrial areas in the northwestern China to the TP.

230 **2.6 LHG**

231 The Qilian Observation and Research Station of Cryosphere and Ecologic Environment,

232 Chinese Academy of Sciences (LHG; 39.50°N, 96.51°E; 4180 m a.s.l.) locates near the

233 terminus (~1 km) of the Laohugou Glacier No.12, which is one the largest mountain

234 glacier in the northern slope of the western Qilian Mountains. LHG is another

235 representative station in the northeastern TP and significantly isolated from the human

236 living areas. The climate in this region is a typically arid and continental climate and

237 dominated by the East-Asian monsoon in the summer and Westerlies in the winter.

238 Strongly mountain-valley breeze is formed during summer which can bring air mass

239 from lower altitude zones to mountain regions. Therefore, it is also well suited for

240 sampling the background air mass and studying the transport mechanisms and potential

241 impacts of air pollutants from its surrounding regions.

242 **2.7 Bayanbulak**

243 The Bayanbulak National Basic Meteorological Station (Bayanbulak; 84.35° N, 42.83°

244 E; 2454 m a.s.l.) locates in the Bayanbulak grassland at the northwest of Hejing county,

245 Xinjiang Uygur Autonomous Region, China. The Bayanbulak is situated in an

246 intermontane basin in the central Tianshan Mountains and surrounded by numerous

247 snow mountains with altitudes more than 3000 m. The climate in Bayanbulak grassland

248 belongs to the typical temperate continental mountain climate with an annual average

249 precipitation of about 200−300 mm. The Bayanbulak town has limited human activities

250 and traffic transportation.

251 **2.8 Lhasa**

252 Lhasa (29.65°N, 91.03°E; 3650 m a.s.l.) is the capital city of the Tibet Autonomous

253 Region, China that located in the south-central part of the TP. The city lies in a flat river

254 valley with the surrounding mountains reaching 5500 m and the Lhasa River passing

255 through the city from west to east. The observation site in Lhasa is in the Binhe Park



near the Lhasa River. The Norbulingka scenic area, one of the main activity centers for
local Tibetans to celebrate their religious festivals (e.g., the Sho Dun festival), is located
~1 km to the northwest of the sampling site, while the Potala Palace, the center of
Tibetan Buddhism, is ~1.8 km to the northeast. Since the unique energy structure and
different residential habit in Lhasa comparing with those remote sites, a comparative
observation is conducted in this urban site for studying the primary aerosol properties
and sources from various residential combustion activities.

## 3 Online sampling, instrumental setup, and data processing

### 3.1 Online real-time aerosol sampling over the TP

The online-based observations of atmospheric aerosols were carried out at each site
using a suit of real-time high-resolution instruments, usually including a HR-ToF-AMS
(Aerodyne Research Inc., Billerica, MA, USA) for acquiring the chemical compositions
(organic aerosol (OA), nitrate, sulfate, ammonium, and chloride) of non-refractory
submicron aerosol ($PM_1$), a scanning mobility particle sizer (SMPS, model 3936, TSI
Inc., Shoreview, MN, USA) for acquiring the size distribution of number concentration
of submicron particles, a photoacoustic extinctiometer (PAX, DMT Inc., Boulder, CO,
USA) for acquiring the particle light absorption, scattering, and extinction coefficients
($B_{abs}$, $B_{scat}$, and $B_{ext}$) and single scattering albedo (SSA) at 405 nm as well as the mass
concentration of BC, an Aethalometer (model AE33/AE31, Magee Scientific Corp.,
Berkeley, CA, USA) for acquiring the $B_{abs}$ at seven fixed wavelengths (370−950 nm),
and a cloud condensation nuclei (CCN) counter (model CCN-100, DMT Inc., Boulder,
CO, USA) for acquiring the CCN number concentration at different supersaturation (SS)
of water vapor. The specific instrumental deployment and sampling period during each
observation campaign is summarized in Table 1.
The HR-ToF-AMS, serving as the primary instrument for observing atmospheric
aerosol chemistry, was deployed throughout all field campaigns. The SMPS was
utilized at QOMS, Motuo, LHG, and Lhasa, while the PAX was employed at QOMS,
Motuo, Ngari, Waliguan, and Lhasa. Additionally, the Aethalometer was utilized at
QOMS, NamCo, and Waliguan, and the CCN-100 at Motuo, Waliguan, and LHG. Field
observations in the southern, western, or central TP regions were predominantly
conducted during the pre-monsoon periods. For instance, observations took place from
12 April to 12 May at QOMS, 26 March to 22 May at Motuo, 31 May to 1 July at
NamCo, and 1 Jun to 5 July at Ngari, which were to investigate the transboundary



transport of atmospheric pollutants from polluted regions in South Asia to the inland
areas of the TP, considering the influences of Westerlies and the Indian monsoon. On
the other hand, measurements in the remote regions of the northern TP and its
surroundings were carried out during the summer. Specifically, observations occurred
from 1 to 31 July at Waliguan, 4 to 29 August at LHG, and 29 August to 26 September
at Bayanbulak. These measurements were undertaken to monitor the aerosols
transported from surrounding polluted regions, considering the influences of Westerlies
or the intensified East Asian monsoon. The observation in Lhasa was conducted
between 31 August and 26 September focusing on the strongest atmospheric oxidation
capability during the summer season.

**3.2 Instrumental setup**

Despite variations in the instruments used during different observation campaigns, the
sampling settings remain generally consistent. Figure 1b illustrates the fundamental
sampling setup employed for each campaign. All instruments are typically housed
within an air-conditioned trailer or room. The inlets are induced to the instruments from
the roof with a fine particle cyclone (model URG-2000-30EH, URG Corp., Chapel Hill,
NC, USA) in the front of the inlet to eliminate particles with an aerodynamic diameter
($D_{va}$) exceeding 2.5 µm. Subsequently, the fine particles are directed into a Nafion dryer
via 1/2-inch stainless steel tubes to remove moisture from the airflows. Lastly, the
particles are sampled into a series of online instruments for real-time measurements.
For a more comprehensive understanding of the instrumental setups, please refer to our
previous publications (Xu et al., 2018; Zhang et al., 2018; Zhang et al., 2019; Zhao et
al., 2022).

**3.3 Instrumental operation and data processing**

The measurement principles, operation procedures, calibration methods, and data
analysis for these instruments are extensively described in detail in Text S1−4 in the
supplementary materials of this study. Only a few essential descriptions and important
settings are emphasized here as follows: (1) The HR-ToF-AMS instruments were
operated at the V-mode during almost all the eight field campaigns, considering the
relatively low aerosol mass levels and signal-to-noise ratios over the TP regions. (2)
Due to chopper malfunction in the HR-ToF-AMS, particle size observations were not
conducted during the NamCo and LHG campaigns. (3) Different relative ionization



efficiency values were used for ammonium and sulfate according to the ionization
efficiency calibrations of HR-ToF-AMS in different campaigns, while default values
were used to rest species. (4) Different size parameters were achieved according to the
particle sizing calibrations in different campaigns, which ultimately resulted in the
distinct size ranges of non-refractory $PM_1$. (5) Elemental ratios of OA, i.e., oxygen-to-
carbon (O/C), hydrogen-to-carbon (H/C), organic matter-to-organic carbon (OM/OC),
and nitrogen-to-carbon (N/C), were determined using the improved method
(Canagaratna et al., 2015). (6) Default collection efficiency (CE) values of 0.5 were
employed to the HR-ToF-AMS measurements during the QOMS, NamCo, Ngari,
Waliguan, and Lhasa campaigns in consideration of their overall neutralized bulk
submicron aerosols, whereas the composition-dependent CE (Middlebrook et al., 2012)
values are adopted at Motuo, LHG, and Bayanbulak, where bulk submicron aerosols
were slightly acidic. (7) Source apportionments of OA during all observations were
performed by the positive matrix factorization (PMF) analysis method. (8) Only the
chemical compositions of non-refractory $PM_1$ are reported during the Bayanbulak
campaign due to the absence of BC observations. (9) The sample and sheath flow rates
of SMPS were set at 0.3 and 3.0 L $min^{-1}$, respectively, at both QOMS and Lhasa,
measuring particles between 14.6 and 661.2 nm in mobility diameter ($D_m$), whereas 0.5
and 5.0 L $min^{-1}$ at LHG and Motuo sites with a particle size range of 10.9−495.8 nm in
$D_m$. (10) Aethalometer measurement were corrected for both the filter-based loading
effect and multiple scattering effect. A traditional absorption Ångström exponents
(AAE) method (Zhang et al., 2021) was adopted to quantitatively apportion the total
$B_{abs}$ into two parts from BC and BrC ($B_{abs,BC}$ and $B_{abs,BrC}$). (11) During the Motuo,
Waliguan, and LHG campaigns, the number concentrations of CCN were conducted
consecutively at five different $SS$ values of 0.2%, 0.4%, 0.6%, 0.8%, and 1.0% every 5
minutes.

## 4 Aerosol properties, sources, and radiative forcing over the TP

### 4.1 Mass loading and chemical composition of submicron aerosols

Figure S1 provides an overview of the high-time-resolution temporal variations of $PM_1$
chemical species (OA, nitrate, sulfate, ammonium, chloride, and BC) during the eight
observations in the TP and its surroundings. Generally, the mass concentrations of $PM_1$
and its chemical species varied dynamically, with alternating periods of high and low
mass loading throughout the sampling period of each campaign. Despite differences in





sampling years (2015−2022), seasons (March−September), and altitudes (1350−4730
m a.s.l.) at these sites, the significantly distinct $PM_1$ mass and chemical compositions
can effectively reflect the regional difference in aerosol mass levels, properties, and
sources at different regions. On average, the mass concentration of total $PM_1$ across the
eight campaigns ranged from 1.9 to 9.1 µg m$^{-3}$ (Fig. 2 and Table 2). The highest $PM_1$
mass was observed at Waliguan due to the rapid transport of anthropogenic aerosols
and gaseous pollutants from urban areas in northwestern China. Conversely, the lowest
values were measured at NamCo and Bayanbulak, attributed to their background and
pristine environmental conditions. The average $PM_1$ mass level in the TP and its
surroundings was comparable to values observed at other high-altitude, coastal, forest,
and remote background sites worldwide (0.46−15.1 µg m$^{-3}$; Table 3), but significantly
lower than those observed at densely urban (34.4−71.5 µg m$^{-3}$) and suburban
(21.4−44.9 µg m$^{-3}$) sites in other regions of China (Li et al., 2017), suggesting the
predominantly clean nature of atmosphere condition in the remote high-altitude regions
of the TP.
The chemical compositions of $PM_1$ also exhibited significant regional difference (Fig.
2), indicating distinct aerosol sources across different TP regions. OA and BC together
contributed as high as 64.9−85.7% of the total $PM_1$ mass at the five sites of QOMS,
Motuo, NamCo, Ngari, and Lhasa that are located in the southern, western, or central
TP (Table 2). These high contributions were mainly attributed to the frequent transport
of biomass-burning-related emissions from polluted regions in South and Southeast
Asia to the remote sites of TP during the pre-monsoon season (Bonasoni et al., 2010;
Cong et al., 2015; Zhang et al., 2018) as well as the intense local biomass burning
emissions from religious activities in the urban area of Lhasa (Cui et al., 2018; Zhao et
al., 2022). In contrary, the three inorganic species (sulfate, nitrate, and ammonium;
referred to as SNA) accounted for more than 60% of the total $PM_1$ at three northern
sites (Waliguan, LHG, and Bayanbulak). Among these, sulfate had the most dominant
contributions of SNA (38.1−46.0%), consistent with the results observed at another
high-altitude site in the northeastern TP (Menyuan; 28%) and other rural and remote
sites (19−64%) in East Asia, indicating the regional transported sources (Du et al., 2015).
The high contributions of SNA, particularly the sulfate, in the northern TP and its
surroundings were mainly related to the regional transport of anthropogenic aerosols
and gaseous precursor emissions from surrounding urban areas as well as important in-





cloud aqueous reactions during the transportation to the mountains (Zhang et al., 2019).
**4.2 Bulk acidity, size distribution, and diurnal variation of submicron aerosols**
Particle phase acidity is an important parameter affecting the physicochemical
properties of aerosol, with significant impacts on hygroscopic growth, toxicity, and
heterogeneous reactions of aerosol particles. The bulk acidity of submicron aerosols
was evaluated at each site following the method in Zhang et al. (2007b) and
Schueneman et al. (2021) based on AMS measurements. A detailed description of this
method can be found in our previous publications (Zhang et al., 2018; Zhang et al.,
2019) or in Text S5 in the supplementary materials of this study. It is interesting to find
that the bulk acidity of submicron aerosols exhibited distinct regional difference
between the southern and northern TP (Fig. 3), primarily attributed to variations in
aerosol sources and compositions. Linear regression slopes were fitted to be 1.2, 1.11,
0.98, and 1.18 at the four sites of QOMS, NamCo, Ngari, and Lhasa that locates in the
southern, western, or central TP, indicating the submicron aerosol particles at these sites
were generally neutralized, and in some cases, showed an excess of ammonium. The
finding is consistent with previous observations of high ammonia availability resulted
from agriculture emissions in the South Asia (Van Damme et al., 2015). In addition, as
reported in our previous publications, atmospheric aerosols at QOMS and NamCo were
significantly influenced by the transport of biomass-burning-related emissions from
South and Southeast Asia during the pre-monsoon season (Xu et al., 2018; Zhang et al.,
2018), while various residential biomass fuels were burned intensely during those
frequent religious festivals in urban areas of Lhasa (Zhao et al., 2022). In contrast, the
submicron particles were overall acidic with regression slopes ranging from 0.73 to
0.86 at the remaining four sites, especially in the two northern sites (LHG and
Bayanbulak), where sulfate significantly contributed to the total $PM_1$ (46.0% and
41.6%). Similar observations of acidic submicron aerosol particles have also been
observed at Menyuan and LHG in the northern TP in previous studies (Du et al., 2015;
Xu et al., 2015), mainly related to the transport of the enriched SNA species or their
gaseous precursors from the industrial areas in northwestern China to the remote
regions in the northern TP.
The size distributions of non-refractory $PM_1$ chemical species, obtained from HR-ToF-
AMS measurement, provide valuable insights into aerosol sources, oxidation degrees,
mixing states, formation, transformation, and growth mechanisms as well as their



impacts on CCN activity. Typically, the size distributions peaked at accumulation mode
(~400−600 nm in $D_{va}$) for those SNA species and oxidized OA components as a result
of their secondary formation. In contrast, fresh organics from primary emission sources
had smaller sizes (Zhang et al., 2005b; Aiken et al., 2009). In this study, both organics
and the sum of three SNA species were selected to illustrate the regional difference in
size distributions across the different TP regions. As shown in Figs. 4a and S2 and Table
2, the peak diameters in the size distribution of OA and SNA varied significantly,
ranging from 584.4 and 634.5 nm at Ngari to a smaller size of 228.1 and 250.0 nm at
Lhasa, respectively. This suggests distinctly different sources and aging processes of
atmospheric aerosols in different TP regions, particularly between those high-altitude
remote sites and urban sites. For example, bulk $PM_1$ was reported to be internally well-
mixed and aged at QOMS due to long-range transport aerosol sources of biomass-
burning-related emissions from South Asia (Zhang et al., 2018), whereas local primary
sources including cooking, traffic exhausts, and biomass burning together contributed
more than 60% of the total OA at the urban site in Lhasa (Zhao et al., 2022). The crucial
influence of aerosol sources on size distributions is further supported in Fig. 4a by the
quantitative relationship between the mode size and O/C ratios of OA ($R^2 = 0.74$).
The diurnal variations of $PM_1$ chemical compositions are typically influenced by
multiple factors, including the meteorological conditions (such as planetary boundary
layer (PBL) height, wind direction and speed, temperature, relative humidity), different
primary emission sources (such as intense vehicle exhausts during traffic rush hours,
cooking emissions, and coal combustion emissions from heating activities), and distinct
formation mechanisms (such as daytime photochemical oxidation processes, nighttime
heterogeneous reactions, and gas-particle partitioning of secondary species). Therefore,
a comprehensive understanding of the diurnal variation characteristics of different
aerosol chemical compositions is not only beneficial for investigating their dynamic
evolution processes but also helpful in understanding the key factors (source,
meteorology, or secondary formation) that drive the variations of different chemical
species. Clearly, different diurnal variation patterns of the total $PM_1$ mass
concentrations were observed during the different field campaigns (Fig. 4b). The
diurnal variations at those remote sites (such as QOMS, LHG, NamCo, and Waliguan)
located either in the valley or at the top of the mountains were mostly controlled by the
circulation of mountain-valley wind and the variation of PBL height during the day.





QOMS exhibited a distinct diurnal pattern with continuously decreasing concentrations
during the daytime, but relatively higher concentrations at night. The minimum mass
occurred at around ~15:00 in this valley site, mainly due to the strong down-slope
glacier winds with high wind speed and enhanced PBL height in the afternoon (Zhang
et al., 2018). Conversely, lower $PM_1$ mass concentration from night to early morning
and continuously increasing concentrations during the afternoon were observed at LHG
and NamCo sites. The high mass concentrations in the afternoon at LHG were tightly
associated with the transport of aerosols advected by the prevailing up-slope winds
during that time. However, the high concentration in the afternoon at NamCo might be
influenced by the downward transmission of aerosols from higher layer to the ground
surface, as well as enhanced aerosol plume transport from those relatively polluted
western regions during the afternoon (Xu et al., 2018). A relatively complex diurnal
variation pattern of total $PM_1$ was observed at the top of Mt. Waliguan, which could be
attributed to the combined effects of variabilities in diffusion conditions (such as PBL
height), wind directions (such as mountain-valley wind circulation), and air mass
sources (such as enhanced air mass from the northeast during the afternoon, favoring
the transport of polluted aerosols from industrial areas) (Zhang et al., 2019). At Motuo,
the diurnal variation of $PM_1$ mass concentration was relatively stable, except for two
weak peaks in the late morning and evening, which were possibly related to combustion
emissions from local residents in the county. Ngari exhibited relatively high mass
loadings at night, whereas low values were observed during the daytime, mainly due to
the variations of PBL height. Bayanbulak, on the other hand, had relatively low and
stable $PM_1$ mass throughout the entire day, primarily due to its background feature. In
contrast to the remote sites, the urban site in Lhasa showed a clear diurnal variation
pattern with two peaks around 8:00–9:00 and 20:00–21:00. This pattern could be
attributed to strong primary aerosol emissions from vehicle exhausts, cooking, and
biomass burning activities during the morning and evening times (Zhao et al., 2022).
Although the diurnal variations of total $PM_1$ were mainly affected by the variabilities
in mountain-valley wind circulation and PBL height in those remote sites and primary
emissions in the urban site in this study, the photochemical oxidation and aqueous-
phase reactions were also two important formation pathways of secondary inorganic
and organic aerosol species. This could be observed clearly by those identified
oxygenated OA (OOA) components at almost all the sites, which commonly showed
peaks during the afternoon (Xu et al., 2018; Zhang et al., 2018; Zhang et al., 2019; Zhao



et al., 2022).

**4.3 High-resolution mass spectrum and elemental ratios of organic aerosol**

The high-resolution mass spectrum (HRMS) and elemental ratios of OA were
determined to identify the possible sources, formation and evolution mechanisms, as
well as the oxidation states of these complex OA components at each site. The average
O/C ratios of the OAs from the eight field campaigns were compared directly (Fig. 5a).
It is evident that the average O/C ratios of OAs were generally close to or larger than
1.0 at the remote sites of QOMS, Motuo, NamCo, Ngari, Waliguan, and LHG, whereas
a lower O/C ratio of 0.69 was observed at Bayanbulak and an even lower O/C ratio of
0.44 was observed at the urban site in Lhasa. These differences in O/C ratios were
mainly attributed to the variations in OA sources and aging processes across different
sites. As mentioned earlier, atmospheric aerosols in the remote sites in the TP were
generally associated with long-range transport from surrounding areas, hence the OAs
were generally well-mixed and highly aged during the transport from source region to
the remote sites in the TP (Xu et al., 2018; Zhang et al., 2018; Zhang et al., 2019).
However, local fresh OAs emitted from various residential activities such as cooking,
traffic exhausts, and biomass burning predominated the total OA at urban Lhasa (Zhao
et al., 2022), which ultimately led to a relatively low O/C ratio. Similar differences in
O/C ratios have been observed in previous studies in China. For instance, higher O/C
ratios of 0.98, 1.11, and 1.16 were measured at remote sites in Mt. Wuzhi (Zhu et al.,
2016), Mt. Yulong (Zheng et al., 2017), and LHG (Xu et al., 2015), while O/C ratios of
OAs were generally lower than 0.5 at most urban sites (Zhou et al., 2020). The Van
Krevelen diagram (H/C versus O/C), a widely used approach to depict the changes in
the elemental composition of OA resulting from atmospheric aging processing, is
displayed in Fig. 5b. An overall slope of −0.66 was observed for the bulk OAs across
the eight field measurement campaigns in our study, which is comparable to those of
−0.58 and −0.47 obtained from different synthesized datasets from diverse field
observations in previous studies (Chen et al., 2015; Zhou et al., 2020).
The average HRMSs of OA between the remote site (Waliguan) and the urban site
(Lhasa) were directly compared to investigate the inherent difference in ion
compositions (Fig. 5c). Waliguan was chosen as an example because the overall highly-
aged OA nature with very similar HRMSs among the seven remote sites, as shown in
Fig. S3 in the supplementary materials. It is evident that the OA HRMSs exhibited




distinct variations between these two types of sites. At the Waliguan site, *m/z* 44, which
is composed totally by $CO_2^+$ and one of the most reliable markers for OOA, was the
base peak (18%) in the OA HRMS. The $CO_2^+$ and its related four ions ($CO^+$, $H_2O^+$,
$HO^+$ and $O^+$) together contributed more than 41% of the total OA signals. Additionally,
the two oxygenated ion fragments ($C_xH_yO_1^+$ and $C_xH_yO_2^+$) accounted for as much as
66% of the total OA signals (Fig. 5c). All these features demonstrated the overall highly
oxygenated nature of OA at the remote background sites in the TP. In contrast, the OA
HRMS at Lhasa was remarkably similar to those observed at most urban cities. The
four *m/z*s value at 43, 55, 57, and 60, which are recognized as important mass spectral
tracers for less oxidized OA compounds or primary emissions related to traffic, cooking,
and biomass burning activities (Zhang et al., 2005a; Alfarra et al., 2007; He et al., 2010),
showed significant contributions to the total OA signals in this urban site. Specifically,
the ion fragment of $C_xH_y^+$ contributed as higher as 64.5% of the total OA signal in
Lhasa, whereas the two oxygenated ion fragments contributed only 33.6%. The high
contribution of fresh ion fragments in the OA HRMS in Lhasa is comparable to those
measured at other urban cities, such as 56% and 59% in Lanzhou (Xu et al., 2014b; Xu
et al., 2016), 51.2% in Nanjing (Wang et al., 2016), and 51.2% in New York (Sun et al.,

537 2011).

**4.4 OA components from PMF source apportionment**

Source apportionments of OA were performed using PMF analysis on OA HRMS data
for each field campaign. Figure 6 presents the average mass contributions of different
OA components from the selected 2−4 factor solutions among the eight different field
campaigns, while the specific HRMS of each OA component is displayed in Fig. S4.
Due to the limited local sources but dominated sources from regional transport, only
two secondary OOA factors with different oxidation degrees, namely a less oxidized
OOA (LO-OOA) and a more oxidized OOA (MO-OOA), were identified during the
NamCo, LHG, and Bayanbulak campaigns. On average, during the NamCo campaign,
MO-OOA and LO-OOA, with average O/C ratios of 0.96 and 0.49, respectively,
accounted for 59.0% and 41.0% of the total OA mass. Similarly, the Bayanbulak
campaign exhibited contributions of MO-OOA (average O/C of 1.12) and LO-OOA
(average O/C of 0.55) to total OA mass at 66.3% and 33.7%, respectively. However,
the LHG campaign showed only 24.9% of MO-OOA and 75.1% of LO-OOA with
relatively high O/C ratios of 1.29 and 1.08, respectively. Besides the two OOA factors,



biomass-burning-related OA (BBOA) was also widely identified in the TP regions. At
QOMS, the total OA mass was composed by 42.4% of MO-OOA, 43.9% of BBOA,
and 13.9% of nitrogen-containing OA (NOA), with average O/C ratios of 1.34, 0.85,
and 1.10, respectively. The high O/C ratio and significant contributions from BBOA
and NOA at QOMS were associated with the transport of biomass burning emissions
from polluted regions in South Asia to the Himalaya and inland TP regions during the
pre-monsoon season (Cong et al., 2015; Zhang et al., 2018; Kang et al., 2019a). At
Waliguan, the total OA mass was composed by 34.4% of MO-OOA, 40.4% of relatively
aged BBOA (agBBOA), 18.3% of BBOA, and 6.9% of hydrocarbon-like OA (HOA),
with average O/C ratios of 1.42, 1.02, 0.69, and 0.33, respectively. The two BBOA
components, particularly agBBOA, exhibited an enhanced contribution to total OA as
the OA mass concentration increased, ranging from only ∼10% to as high as 70% when
OA mass varied from <1.0 µg m$^{-3}$ to 7 µg m$^{-3}$ (Zhang et al., 2019). In addition, source
analysis indicated that the high contributions of the two BBOA components at Waliguan
were associated with the regional transport of biomass burning emissions from the
residential areas in northeastern Waliguan (Zhang et al., 2019). At Ngari, the total OA
mass was composed by 43.7% of MO-OOA, 28.5% of LO-OOA, and 27.8% of BBOA,
with average O/C ratios of 1.43, 1.00, and 0.56, respectively. In comparison, the
contributions of the three OA components at Motuo site were 36.9%, 46.9%, and 16.2%,
with O/C ratios of 1.30, 1.11, and 0.25, respectively. The BBOA factor at Motuo
exhibited relatively lower mass contribution and O/C ratio compared to those observed
at QOMS, Ngari, and Waliguan, suggesting weak local source from biomass burning.
Four OA factors including an OOA component with O/C ratio of 0.54 and three primary
OA components, i.e., a BBOA with O/C of 0.13, a cooking-related OA (COA) with O/C
of 0.12, and a HOA with O/C of 0.11, were identified at the urban site in Lhasa, which
were distinctly different with those observed at above remote sites. The three primary
OA components together contributed more than 60% of the total OA mass at Lhasa,
suggesting the abundant primary aerosol sources from the residential activities. In
addition, the BBOA contribution increased obviously (up to 36%) during a grand local
festival at Lhasa, suggesting the crucial aerosol source from biomass burning during
religious activities in the city (Zhao et al., 2022). In summary, distinct types of OA
components with different O/C ratios were identified at different sites, indicating the
different sources and oxidation states of OA in the different TP regions.



### 4.5 Number concentrations of submicron aerosols and cloud condensation nuclei

Real-time online measurements of the size distribution of number concentration of fine particles were also conducted simultaneously using SMPS instruments during four field campaigns (QOMS, Motuo, LHG, and Lhasa). The measurement of particle number size distribution (PNSD) was not only an important auxiliary data for calibrating and verifying the accuracy of HR-ToF-AMS data, but also very useful for studying the formation and growth mechanisms of aerosol particles in the atmosphere. Figure 7a shows the high-resolution temporal variations of the PNSDs during the four field campaigns. The PNSDs varied dynamically throughout the measurement period at each site and showed distinct variations in number concentrations and size distribution pattern among the four different campaigns. On average, the total number concentrations were 709.3 and 3994.4 $cm^{-3}$ at QOMS and Lhasa, respectively, while they were 1639.2 and 1462.0 $cm^{-3}$ at Motuo and LHG. Interestingly, the difference in particle number concentrations were not consistent with those in mass concentrations measured from the HR-ToF-AMS among the four campaigns (Table 2). For example, although the $PM_1$ mass concentration at Lhasa was comparable to that at QOMS (4.7 versus 4.4 $\mu g\ m^{-3}$), the number concentration at Lhasa was more than five times higher than that at QOMS. This inconsistency may be related to the distinctly different size distribution at different sites. As discussed above, submicron aerosols at QOMS were overall highly aged due to the long-range transported sources from South Asia and dominated by aerosols at accumulation size mode, whereas more fresh aerosols emitted from local residential activities and dominated by aerosols of Aitken size mode were observed at Lhasa. The different sizes of submicron aerosols among the different TP regions could be further confirmed by the peak diameters in the average size distributions of mass and number concentrations (Figs. 4a and 7b). For example, the average OA mass size distributions exhibited peak diameters of 510.2 and 430.5 nm in $D_{va}$ at QOMS and Motuo, respectively, while the average number size distributions had peak diameters of 109.4 and 131.0 nm in $D_m$ at the same sites. In contrast, Lhasa displayed peak diameters of only 228.1 nm in $D_{va}$ and 28.9 nm in $D_m$.

New particle formation (NPF) events were also observed at a few sites in our study. Typically, an NPF event is characterized by a rapid burst in nucleation mode followed by the subsequent growth into larger particles, as defined as banana-shaped temporal developments in the PNSD (Dal Maso et al., 2005). As shown in Fig. 7a, frequent





banana-shaped variation patterns in the PNSD were observed at Lhasa, suggesting the frequent occurrence of NPF at this urban region. During the Lhasa campaign (27 days), a total of 10 NPF events were observed (Zhao et al., 2022). In contrast, this banana-shaped pattern was relatively rare at the other three remote sites, which might be related to their predominated transport aerosol sources, overall highly-aged states, and limited gaseous precursors.

Cloud condensation nuclei (CCN) is a distinct class of atmospheric aerosol particles which could be activated as cloud droplets at a certain supersaturated water vapor condition and played important roles in cloud formation, atmospheric precipitation, the regional climate, as well as the hydrological cycle (Andreae and Rosenfeld, 2008). During the TP field campaigns, real-time online measurements of CCN number concentrations were conducted at three sites, i.e., Motuo in the southeastern TP while Waliguan and LHG in the northeastern TP. Generally, the temporal variation of CCN number concentration exhibited a consistent trend with the total number concentration from the SMPS measurement or total $PM_1$ mass concentration from the HR-ToF-AMS measurement during each campaign. On average, at Motuo, the CCN number concentrations were 974.0, 1142.6, 1240.1, 1296.5, and 1337.9 $cm^{-3}$ at different $SS$ values of 0.2%, 0.4%, 0.6%, 0.8%, and 1.0%, respectively. At Waliguan, relatively comparable average values of 507.0, 805.1, 1073.3, 1230.6, and 1336.6 $cm^{-3}$ were observed at the same SS steps, respectively. However, at LHG, these average values significantly decreased to 83.9, 344.3, 429.9, 480.8, and 516.1 $cm^{-3}$, respectively (Table 2). The lower CCN number concentrations at LHG compared to Waliguan and Motuo were consistent with the relatively lower $PM_1$ mass loading at the LHG site. The CCN number concentrations at the three TP sites were almost an order of magnitude lower than those observed in polluted urban atmospheres or emissions of specific combustion sources, such as 12963 $cm^{-3}$ ($SS$ = 0.70%) in Wuqing, 9890 $cm^{-3}$ ($SS$ = 0.86%) in Beijing (Deng et al., 2011; Gunthe et al., 2011), 7913 $cm^{-3}$ ($SS$ = 0.70%) at Panyu in the Pearl River Delta, as well as 11565 $cm^{-3}$ ($SS$ = 0.87%) and 10000 $cm^{-3}$ ($SS$ = 0.80%) during unique biomass burning plumes (Rose et al., 2010; Zhang et al., 2020). However, our values were comparable to those (228−2150 $cm^{-3}$ with $SS$ of 0.87%) measured at eight remote marine sites in the South China Sea and 941 $cm^{-3}$ ($SS$ = 0.74%) in the amazon rain forest (Pöhlker et al., 2016; Atwood et al., 2017). These comparisons again highlight the overall clean atmospheric condition in the TP.



### 4.6 Aerosol optical properties and light absorptions from BC and BrC

The optical properties of aerosol particles are crucial input parameters for accurately
estimating aerosol radiative forcing in climate models. However, significant
uncertainties persist due to the limited dataset in this remote region. In our project, the
parameters of $B_{scat}$, $B_{abs}$, and SSA of fine particles at 405 nm were observed during five
field campaigns, i.e., QOMS, Motuo, Waliguan, Ngari, and Lhasa, to explore the
differences in aerosol optical properties at different TP regions. On average, the $B_{scat}$
and $B_{abs}$ at 405 nm during the five campaigns were 121.9, 44.9, 36.3, 8.9, and 2.1 Mm$^{-1}$
and 10.8, 7.0, 4.1, 3.6, and 1.9 Mm$^{-1}$, respectively, which finally resulted in average
SSA values of 0.89, 0.83, 0.86, 0.67, and 0.52, correspondingly (Fig. 8a and Table 2).
These $B_{scat}$ and $B_{abs}$ values at the five TP sites were both significantly lower than those
reported at various urban sites in China, such as 459.5 and 47.2 Mm$^{-1}$, respectively, at
630 nm in Beijing (Xie et al., 2019), 272 and 31 Mm$^{-1}$, respectively, at 532 nm in Xi'an
(Zhu et al., 2015), and 418 and 91 Mm$^{-1}$, respectively, at 540 nm in Guangzhou
(Andreae et al., 2008), once again suggesting the overall clean atmospheric condition
in the TP. Although the PM$_1$ mass at QOMS was comparable to or lower than those at
the other four sites, the highest $B_{scat}$, $B_{abs}$, and SSA values were observed at QOMS.
These results may be attributed to the differences in aerosol chemical compositions and
mass scattering and absorbing efficiencies. In contrast, Lhasa exhibited a significantly
lower SSA compared to the other four remote sites, suggesting a prevalence of fresh
aerosols in the urban area. On the other hand, aerosols at the four remote sites were
highly aged, which resulted in significant photobleaching in BrC chromophores and an
obvious decrease in their light absorptivity at these sites.
In this study, real-time online measurements of particle $B_{abs}$ at seven fixed wavelengths
(370−950 nm) were also conducted using an aethalometer at QOMS, NamCo, and
Waliguan, respectively, to explore the regional difference in aerosol absorption
properties in the different TP regions. Overall, the muti-wavelength $B_{abs}$ decreased
significantly with the increasing wavelength during all the three measurement
campaigns, with fitting AAE values to be 1.73, 1.28, and 1.12, respectively (Fig. 8b).
The average $B_{abs}$ at the shortest wavelength of 370 nm was 13.40, 3.25, and 2.66 Mm$^{-1}$
at the three sites, respectively (Table 2). Although a relatively low PM$_1$ mass was
observed at QOMS, the $B_{abs}$ at 370 nm was five times higher than that at Waliguan,
mainly as a result of the higher contribution of light-absorbing aerosol components in



the southern TP regions. For example, OA and BC together contributed nearly 80% of
the total $PM_1$ at QOMS, whereas this contribution decreased to only 37.5% at Waliguan.
The obviously higher AAE at QOMS also suggested a dominant light-absorbing
contribution from BrC or the significant lensing effect of non-BC materials coated on
BC at this southern site (Zhang et al., 2021). As shown in the inserted plots in Fig. 8b,
both BC and BrC components showed significant decrease of particle $B_{abs}$ ($B_{abs,BC}$ and
$B_{abs,BrC}$) with increasing wavelength, but their contributions to total $B_{abs}$ ($fB_{abs,BC}$ and
$fB_{abs,BrC}$) varied inversely. BC was the primary light-absorbing component at all the
three sites, contributing 66.9%, 78.7%, and 77.6% to the total $B_{abs}$ at 370 nm at QOMS,
NamCo, and Waliguan sites, respectively, and its contribution increased apparently with
longer wavelengths (Table 2). BrC showed more significant contributions to total $B_{abs}$
at shorter wavelengths. For example, the average $B_{abs,BrC}$ at 370 nm were 4.42, 0.69,
and 0.60 $Mm^{-1}$ at the three sites, respectively, which finally contributed 33.1%, 21.3%,
and 22.4% of the total $B_{abs}$. The significantly higher values of total $B_{abs}$, $B_{abs,BC}$, $B_{abs,BrC}$,
and $fB_{abs,BrC}$ in the southern TP region could be related to the important contributions
of light-absorbing CAs from transported biomass burning emissions (Xu et al., 2020,
701 2022).

**4.7 Estimation of aerosol radiative forcing in the different TP regions**
Atmospheric aerosols have been found to significantly impact the Earth's climate
systems through affecting solar radiation and exerting a positive forcing on the energy
budget (Bond and Bergstrom, 2006). In this study, aerosol direct radiative forcings
(DRF) caused by BC, organic carbon (OC), and water-soluble ions (WSIs) are
estimated, respectively, by the widely used Santa Barbara DISORT (Discrete Ordinate
Radiative Transfer) Atmospheric Radiative Transfer (SBDART) model (Ricchiazzi et
al., 1998). A detailed introduction and operation process of this model are described in
Text S6 in the supplementary material. Since the model's performance is evaluated and
calibrated by comparing the values with measurements from the Aethalometer and PAX
instruments, the aerosol DRF estimations are limited to the three sites of QOMS,
NamCo, and Waliguan, which have both online measurements from the aforementioned
instruments. Furthermore, these three sites are located in the southern, central, and
northern regions of the TP, respectively, which enables us to explore the regional
variations in aerosol DRF across different TP regions.
Figure 9 presents the modelled DRFs caused by BC, OC, and WSIs during the three





campaigns. BC exhibited a significant warming effect at the top of the atmosphere, with
average DRF values of $+2.5 \pm 0.5$, $+2.1 \pm 0.1$, and $+1.9 \pm 0.1$ W m$^{-2}$ during the QOMS,
Waliguan, and NamCo campaign, respectively. In contrast, a noticeable cooling effect
caused by BC was observed at the earth's surface with average DRF values of $-4.7 \pm$
$0.8$, $-4.1 \pm 0.2$, and $-3.7 \pm 0.1$ W m$^{-2}$ across the three campaigns. The combination of
warming effect at the top of the atmosphere and cooling effect at the earth's surface
resulted in significantly high net atmospheric forcings by BC, amounting to $+7.3 \pm 1.2$,
$+6.2 \pm 0.3$, and $+5.6 \pm 0.2$ W m$^{-2}$ during the QOMS, Waliguan, and NamCo campaigns,
respectively. These findings suggest the important radiative effect caused by BC in the
TP, especially in the southern TP region, which was significantly influenced by the
long-range transported biomass burning emission from South Asia. For OC and WSIs,
cooling effects were observed at both the top of the atmosphere and the earth's surface,
characterized by negative and low average DRFs. Consequently, the net atmospheric
forcings for OC and WSIs were significantly lower compared to BC across the three
campaigns, with values of $+2.0 \pm 1.2$, $+0.7 \pm 0.2$, and $+0.9 \pm 0.7$ W m$^{-2}$ for OC, and
$+1.9 \pm 0.8$, $+1.4 \pm 0.6$, and $+1.2 \pm 0.2$ W m$^{-2}$ for WSIs at QOMS, Waliguan and NamCo,
respectively. Interestingly, at QOMS, the average atmospheric DRF of OC accounted
for 27.3% of that of BC, whereas at Waliguan and NamCo, the fractions were only 11.1%
and 15.7%, respectively. The higher atmospheric DRF observed at QOMS suggests a
dominant contribution from light-absorbing BC and BrC aerosols, compared to
Waliguan and NamCo. It was worth noting that the simulations of DRF effects in this
study were only conducted at three specific sites during limited periods. Therefore,
long-term comprehensive measurements and DRF simulations over the entire TP
regions under different seasons are needed in the future.
**4.8 Long-range transport of aerosols from surrounding areas**
To further understand the potential sources and specific transport pathways of air
masses at each site, particularly for those remote sites where regional transport
dominated, three- or five-days air mass back trajectories were calculated during each
measurement period at an ending height of 500 m above ground level every 6h using
the Hybrid Single Particle Lagrangian Integrated Trajectory (HYSPLIT) model
(Draxler and Rolph, 2003). Figure 10 display the average backward trajectory clusters
during all the eight field campaigns and the major trajectory clusters at each site are
marked with large solid circles in different colors.



In general, distinct air mass sources were identified among the different TP regions. The
five sites (QOMS, Motuo, Lhasa, NamCo, and Ngari) located in the southern or south-
central part of the TP generally showed dominant air mass sources from the south or
southwest with different transport distances and pathways during their measurement
periods in pre-monsoon season. For example, during the QOMS campaign, 38% of the
air masses originated from the west, covering a considerably long transport distance,
while another 40% was originated from the southwest, covering a relatively shorter
distance. In the Motuo campaign, two major clusters were both from the southwest, but
their transport distances were distinctly different (77% at shorter distance compared to
only 13% at a longer distance). Similarly, during the NamCo campaign, two different
major clusters with comparable contributions (37% and 34%) and transport distances,
but different transport pathways, were found from the south. The Ngari campaign also
observed similar transport distances, with 56% of the air masses originating from
southwest and 26% from south of the site. These air mass clusters originating from the
south of the TP generally traverse heavy polluted regions in South Asia, such as the
Indo-Gangetic Plain, Nepal, and Bangladesh, carrying significant amounts of polluted
aerosols, particularly the biomass-burning-related emissions from the source origins to
the inland of the TP. In contrast, air masses at the northern sites were primarily
influenced by the prevailing Westerlies wind and East Asian monsoon during the
summer season measurement periods. In the Bayanbulak campaign, the major air
masses were all originated from the west of the site with varying transport distances
(i.e., 69% in relatively shorter distance versus 18% in a longer distance). During the
LHG campaign, the air masses originated from the northwest of the site with 63% in
longer transport distance but 27% in shorter distance. For the Waliguan campaign, the
air mass clusters originated from two distinctly different directions. The majority of the
air masses (57%) came from the northeast of the site covering a relatively shorter
distance and faster transport speed, while the remaining clusters originated from the
west and northwest of the site covering significantly longer distances. In summary,
significant differences in air mass sources and transport pathways were identified
among the different TP regions, particularly between the southern and the northern TP
regions. These differences are the primary factors contributing to the significantly
different physiochemical and optical properties of aerosols in the different TP regions.
**5 Dataset limitations and applications**



Our dataset was achieved from multiple short-term intensive field observations
conducted at eight different sites of the TP during their high-mass-loading periods
utilizing a suite of high-resolution online instruments. However, it is important to
acknowledge that our dataset has certain limitations due to objective restrictions that
were quite challenging to overcome in these remote regions.
The primary limitation revolves around the short and inconsistent measurement periods
across different observational years and seasons at different sites. This limitation
hinders the ability to make a robust comparison of aerosol properties across the vast TP
region, ascertain long-term and seasonal variation characteristics, and apply the current
data and findings to other different seasons. The harsh natural environments,
challenging weather conditions, limited logistical support, sole availability of our high-
resolution instruments, and the stringent instrumental requirements (e.g., the need for
comprehensive field stations with uninterrupted and stable power supply) were the most
significant challenges we faced during our field observations in these remote TP regions.
It is worth noting that online HR-ToF-AMS observations, such as the ones we
conducted, are predominantly short-term and intense observations carried out
worldwide due to the stability and its challenging maintenance during long-term
measurement. The short-term intensive measurement can well capture and characterize
the dynamic evolution of aerosol properties at a high-time-resolution (Jimenez et al.,
2009; Li et al., 2017). Until now, long-term high-time-resolution observation utilizing
HR-ToF-AMS have rarely been conducted, even at urban sites with relatively favorable
observational environments and logistic support compared to our remote TP sites.
Consequently, performing continuous long-term observations or simultaneous
comparison at multiple sites in these high-altitude remote and challenging TP regions,
without stable power supply, is nearly impossible.
However, our team has made significant efforts to conduct the comprehensive
observation project over the past ten years, aiming to study the regional differences in
aerosol sources and properties across the different TP regions. The dataset generated
from our project represents the first and sole high-time-resolution dataset focusing on
atmospheric aerosol physicochemical and optical properties, coring the most part of the
TP. The applications of this dataset in atmospheric science can be summarized as
follows: firstly, the high-time-resolution observations offer crucial advantages
understanding the rapid evolution and diurnal variations of aerosol properties during a



short period or special event. Additionally, these observations are valuable for model
simulation and verification, as they provide sufficient data points. Such advantages are
not achievable with traditional off-line samplings, which have low time resolutions
ranging from days to weeks. Secondly, the eight sites included in our project effectively
represent a wide range of the TP. This is particularly noteworthy considering the limited
availability of observational stations on the plateau. Furthermore, these sites provide
excellent opportunities for comparing aerosol sources and properties among different
types of sites with varying altitudes, land covers, surrounding environments, human
activities, and influences from large-scale atmospheric circulations. Thirdly, our
observations encompass a wide range of aerosol physical, chemical, and optical
parameters, including aerosol mass loadings, chemical compositions, size distribution,
diurnal variations, number concentrations, light scattering and absorption coefficients,
and so on. This comprehensive dataset is crucial for a thorough understanding of aerosol
properties in the TP regions. Overall, it is worth noting that until now, similar online
observational aerosol datasets focusing on multiple parameters with at least hourly-
scale resolution at various sites in the diverse TP regions had been rarely reported.

## 6 Data availability

The high-resolution online measurement datasets, encompassing aerosol physical,
chemical, and optical properties over the Tibetan Plateau and its surroundings in our
observation project have been released and are now available for download from the
National Cryosphere Desert Data Center
(https://doi.org/10.12072/ncdc.NIEER.db2200.2022). These datasets are provided in an
Excel file comprising eight worksheets. The first sheet of the Excel file contains a
concise description of the dataset, including the dataset name, observation stations,
sampling periods, online instruments used, and corresponding references. The
remaining seven sheets present the high-resolution measurement data obtained from the
online instruments employed during the eight campaigns. These instruments include
HR-ToF-AMS, SMPS, PAX, aethalometer, and CCN-100.

## 7 Conclusions

A comprehensive dataset including aerosol physicochemical and optical properties,
especially the high-resolution size-resolved chemical characteristics and sources of
submicron aerosols, conducted through real-time online measurements at different sites



of the TP and its surroundings is presented in this study. The objective of this study is
to elucidate the mass concentration level of atmospheric aerosols in this isolated
background region and identify regional variations in aerosol sources, as well as
physicochemical and optical properties among different TP regions. Ultimately, these
valuable data will significantly contribute to accurately simulating the radiative forcing
and other potential impacts of atmospheric aerosols in this remote region in future
climatic models.
A total of eight aerosol field measurements were conducted at QOMS, Motuo, NamCo,
Ngari, Waliguan, LHG, Bayanbulak, and Lhasa in the different regions of TP and its
surroundings by deploying multiple online instruments, including HR-ToF-AMS,
SMPS, PAX, Aethalometer, and CCN-100. The collected datasets provide the temporal
and diurnal variations as well as the size distribution patterns of $PM_1$ chemical
compositions, the standard high-resolution mass spectra and temporal variations of OA
components, the temporal variations of particle number size distribution, particle light
scattering and absorption coefficients, particle light absorptions from different CAs of
BC and BrC, and CCN number concentrations at different supersaturation in each
campaign.
The datasets provide valuable insights into the regional variations in aerosol properties
and sources. In the southern TP region, atmospheric aerosols were found to be primarily
influenced by biomass burning emissions from polluted regions in South Asia, which
resulted in high mass contributions (>70%) of CAs and overall neutralized $PM_1$, as well
as an enhanced light absorption capability of the light-absorbing BC and BrC. In
contrast, in the northern TP, secondary inorganic species, particularly sulfate,
contributed significantly to total $PM_1$ due to the regional transport of anthropogenic
aerosol and gaseous precursor emissions from urban areas in northwestern China.
Furthermore, in contrast to the well-mixed, highly-aged, and regionally transported
aerosols observed in the remote sites, atmospheric aerosols in the urban Lhasa were
mainly originated from local primary sources such as cooking, traffic vehicle exhausts,
and biofuel combustion during the residential activities. Consequently, these aerosol
particles were relatively fresh, characterized by small size and low oxidation degree,
but exhibited a high frequency of NPF origins.
**Appendix A: Main Abbreviations**



| | |
|---|---|
| TP | Tibetan Plateau |
| HR-ToF-AMS | high-resolution time-of-flight aerosol mass spectrometer |
| SMPS | scanning mobility particle sizer |
| PAX | photo-acoustic extinctiometer |
| CCN | cloud condensation nuclei |
| *SS* | supersaturation |
| $PM_1$ | submicron aerosol |
| BC | black carbon |
| BrC | brown carbon |
| OA | organic aerosol |
| SNA | sulfate, nitrate, and ammonium |
| $D_m$ | mobility diameter |
| $D_{va}$ | aerodynamic diameter |
| CE | collection efficiency |
| HRMS | high-resolution mass spectrum |
| PBL | planetary boundary layer |
| O/C | oxygen-to-carbon ratio |
| H/C | hydrogen-to-carbon ratio |
| N/C | nitrogen-to-carbon ratio |
| OM/OC | organic matter-to-organic carbon ratio |
| PMF | positive matrix factorization |
| OOA | oxygenated organic aerosol |
| LO-OOA | less oxidized oxygenated organic aerosol |
| MO-OOA | more oxidized oxygenated organic aerosol |
| BBOA | biomass-burning-related organic aerosol |
| agBBOA | aged biomass-burning-related organic aerosol |
| NOA | nitrogen-containing organic aerosol |
| HOA | traffic-related hydrocarbon-like organic aerosol |
| COA | cooking-related organic aerosol |
| PNSD | particle number size distribution |
| NPF | new particle formation |
| $B_{scat}$ | light scattering coefficient |
| $B_{abs}$ | light absorption coefficient |
| $B_{ext}$ | light extinction coefficient |
| SSA | single scattering albedo |
| AAE | absorption Ångström exponents |
| $B_{abs,BC}$ | light absorption coefficient from BC |
| $B_{abs,BrC}$ | light absorption coefficient from BrC |
| OC | organic carbon |
| WSIs | water-soluble ions |
| DRF | direct radiative forcing |

**Author Contributions.** JX designed the study, XZ, WZ, and JX wrote the manuscript.
JX and SK organized and supervised the field measurement campaigns, JX, XZ, WZ,
LZ, MZ, JS, JShi, YL, CX, YT, KL, XG, and QZ conducted the field measurements,
JX, XZ, WZ, and YT analyzed the data. All authors reviewed and commented on the
final form of the manuscript.



**Competing interests.** The authors declared that they have no competing interests.
**Acknowledgements.** We appreciate all our colleagues and collaborators who
participated the aerosol field measurements, maintained the instruments, analyzed the
data, and commented on the manuscript. We also show great thanks to all the
observation stations in this study for their logistical supports with the field campaigns.
**Financial support.** This work was supported by the National Natural Science
Foundation of China (41977189, 41771079, 41805106), the Second Tibetan Plateau
Scientific Expedition and Research program (STEP) (2019QZKK0605), the Strategic
Priority Research Program of Chinese Academy of Sciences, Pan-Third Pole
Environment Study for a Green Silk Road (Pan-TPE) (XDA20040501), the State Key
Laboratory of Cryospheric Sciences Scientific Research Foundation (SKLCS-ZZ-
2023), and the Chinese Academy of Sciences Hundred Talents Program.

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

# Figures

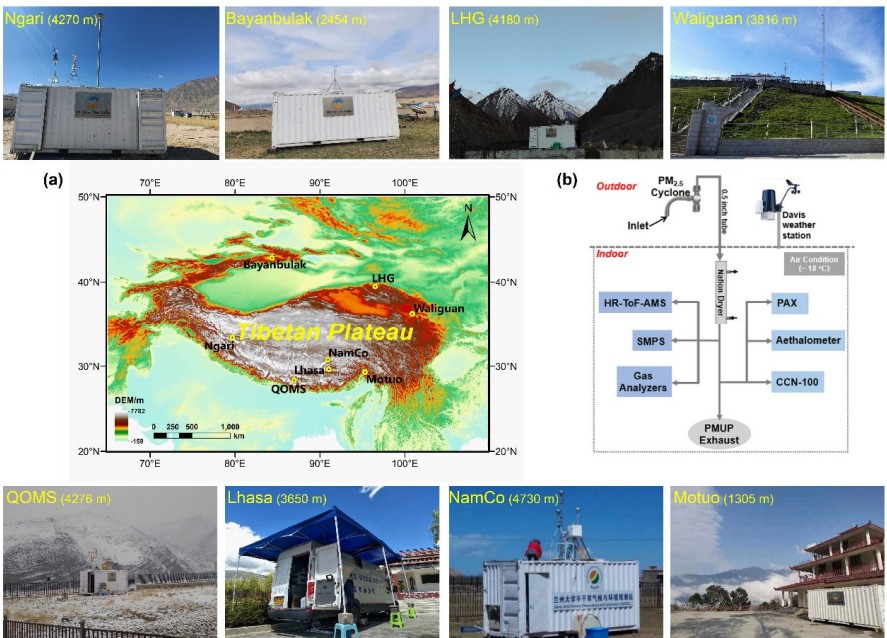

**Figure 1. (a)** Geographical locations of the observation sites (see Table 1 for full name and characteristics of each site) in the Tibetan Plateau and its surroundings in this study (The geographical base map is created with ArcGIS). Fieldwork photographs illustrate the real observation conditions and surroundings at each site. **(b)** The normal sampling setups of instruments during the online aerosol observations.

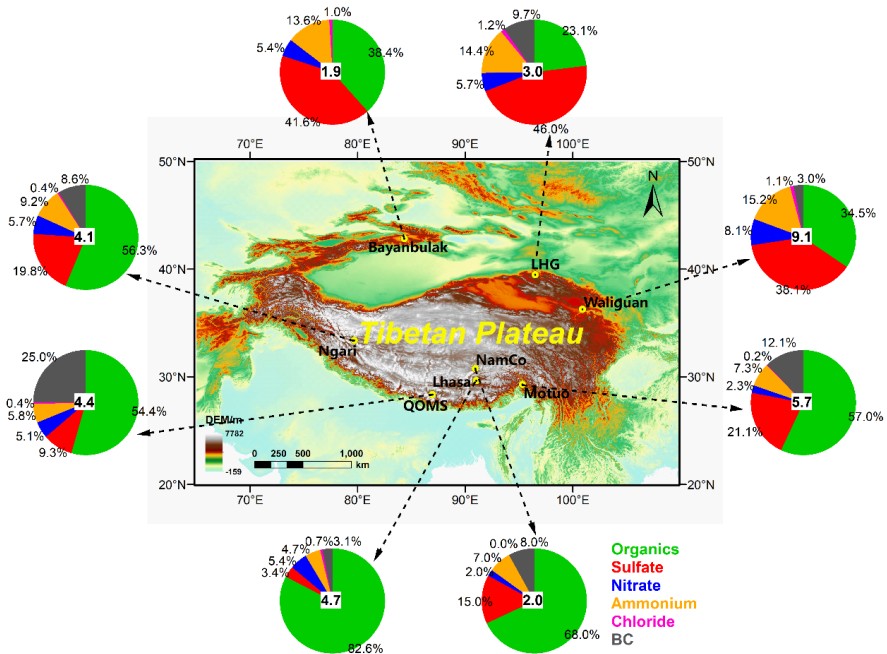

**Figure 2.** Regional distribution of average mass concentrations (values marked in the central of each pie chart with unit of μg m$^{-3}$) and chemical compositions (percentage values around each pie chart) of submicron aerosols (PM$_1$) during the eight online aerosol field measurements in the Tibetan Plateau and its surroundings (The geographical base map is created with ArcGIS).

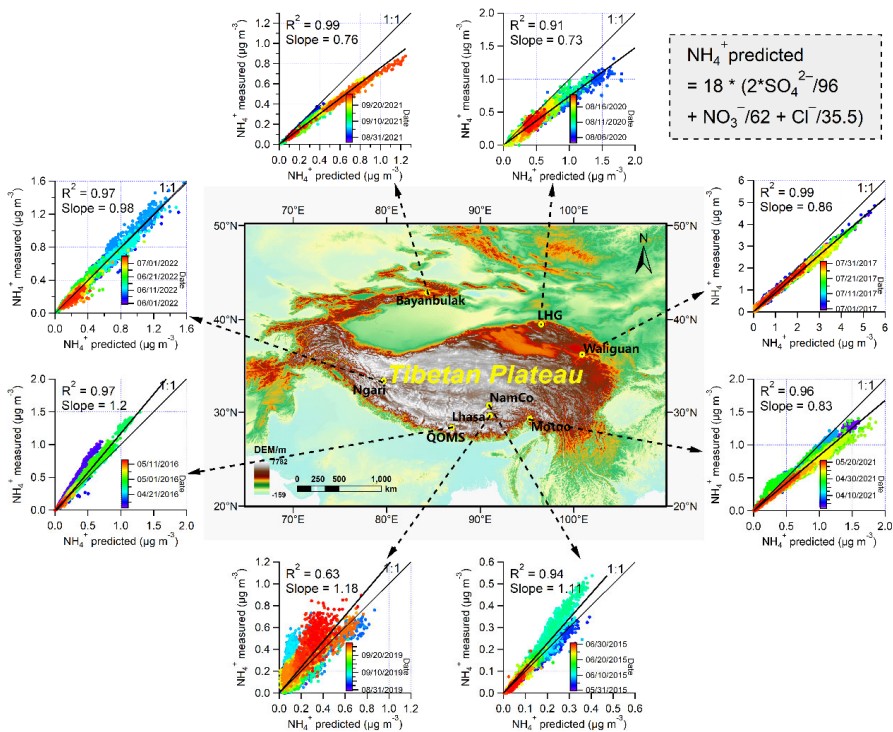

**Figure 3.** Regional difference of bulk acidity of submicron aerosols based on the scatterplot analysis and linear regression of measured NH₄⁺ versus predicted NH₄⁺ during the eight aerosol field measurement campaigns in the Tibetan Plateau and its surroundings (The geographical base map is created with ArcGIS).

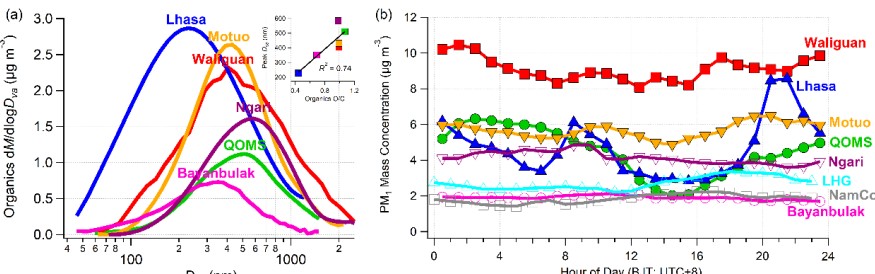

**Figure 4. (a)** Average size distributions of organic mass concentrations during six field measurement campaigns in the Tibetan Plateau and its surroundings. **(b)** Diurnal variations of total $PM_1$ mass concentrations during the eight field measurement campaigns in the Tibetan Plateau and its surroundings. Insert graph in **(a)** is the scatter plot of peak diameters in these size distributions versus the average O/C ratio of organics.

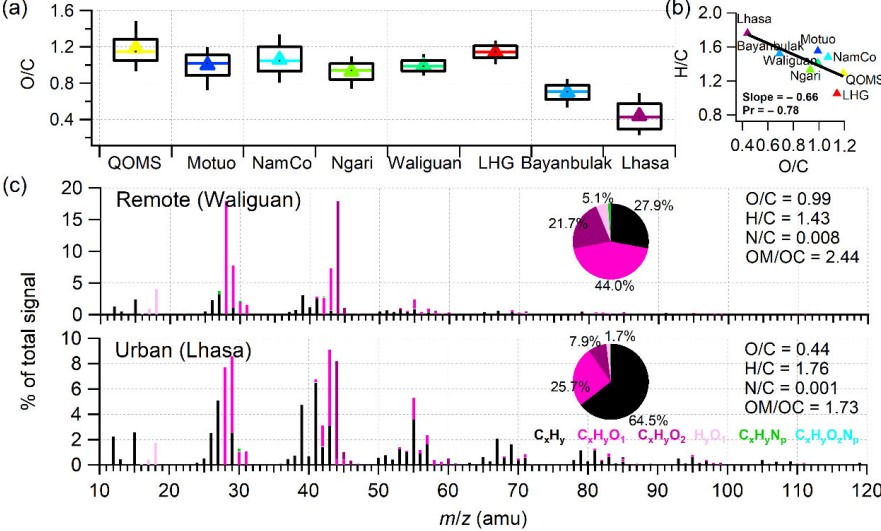

**Figure 5. (a)** Box plots of the average O/C ratios and **(b)** Van Krevelen diagram of H/C versus O/C among the eight field measurement campaigns in this study. **(c)** The average HRMSs of OA colored with different ion categories during the Waliguan and Lhasa measurement campaigns. The whiskers of boxes indicate the 90th and 10th percentiles, the upper and lower boundaries of boxes indicate the 75th and 25th percentiles, the lines in the boxes indicate the median values, the markers indicate the mean values, and similarly hereinafter.

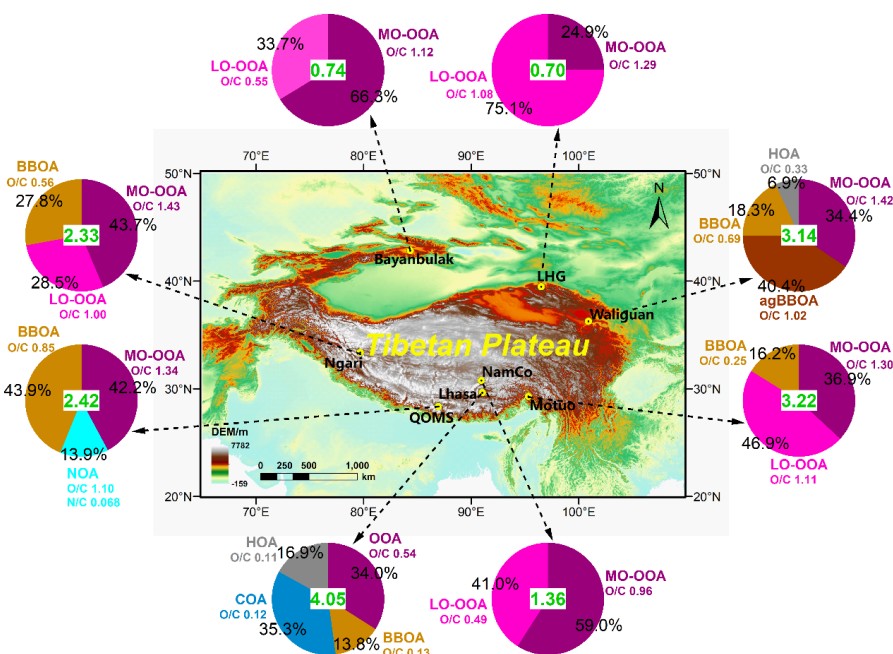

**Figure 6.** Regional distribution of OA components from PMF source apportionment during the eight online aerosol field measurements in the Tibetan Plateau and its surroundings (The geographical base map is created with ArcGIS). Values marked in the central of each pie chart are average OA mass with unit of μg m$^{-3}$ while the percentage values around the pie chart are the mass contributions of each OA component. The O/C ratio of each OA component is also marked around each pie chart.

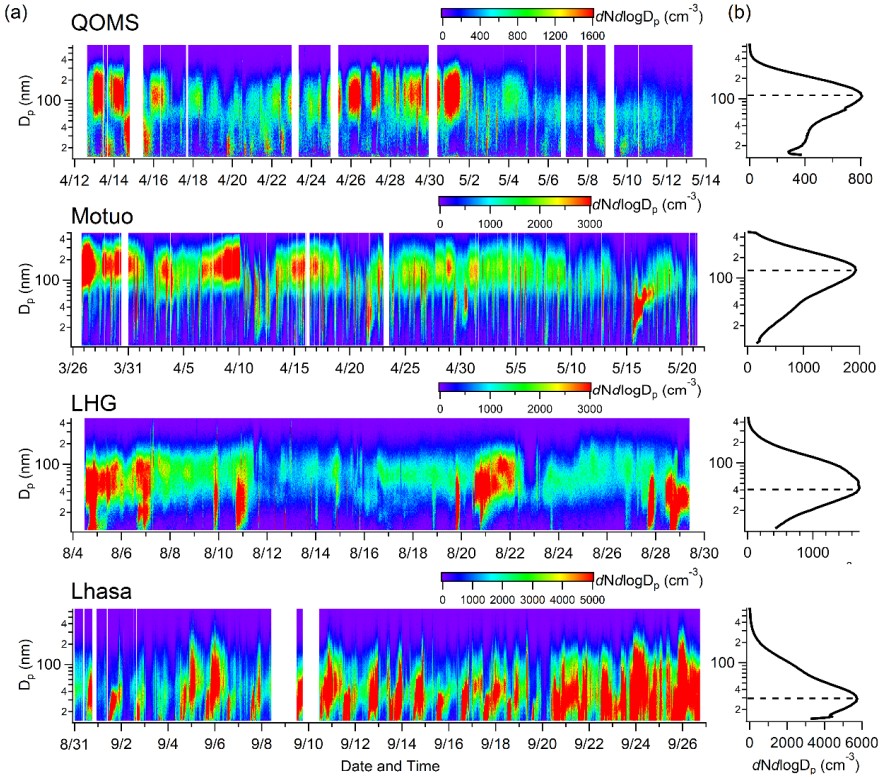

1252

**Figure 7. (a)** Temporal variations of the size distributions of particle number concentrations during the aerosol field measurement campaigns at QOMS, Motuo, LHG, and Lhasa sites. **(b)** The average size distribution of particle number concentration during entire measurement period at each site.



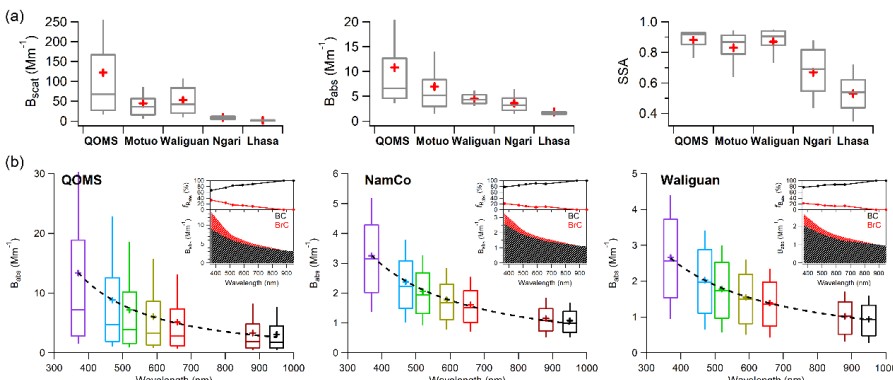

**Figure 8.** Box plots of **(a)** the average particle light scattering coefficient ($B_{scat}$), light absorption coefficient ($B_{abs}$), and single scattering albedo (SSA) during the five aerosol field measurement campaigns at QOMS, Motuo, Waliguan, Ngari, and Lhasa sites, and **(b)** the particle $B_{abs}$ at seven wavelengths measured by aethalometers at QOMS, NamCo, and Waliguan sites. The dashed lines in the boxes in **(b)** show the power-law fit of the average $B_{abs}$ as a function of wavelength. The inserted plots in **(b)** are the apportioned contributions of BC and BrC to total $B_{abs}$ at different wavelengths.

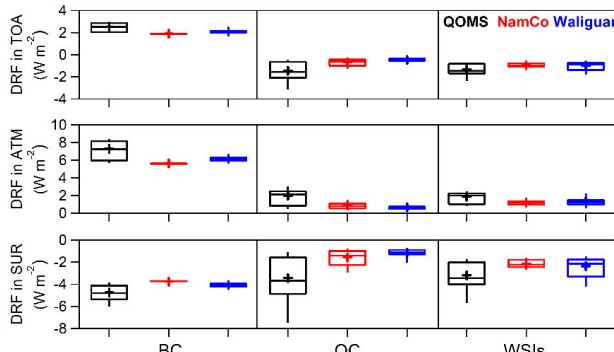

**Figure 9.** Box-plots of the modelled direct radiative forcing (DRF) at the top of the atmosphere (TOA), the atmosphere (ATM), and the earth's surface (SUR) caused by black carbon (BC), organic carbon (OC), and water-soluble io ns (WSIs) during the QOMS, NamCo, and Waliguan campaigns.

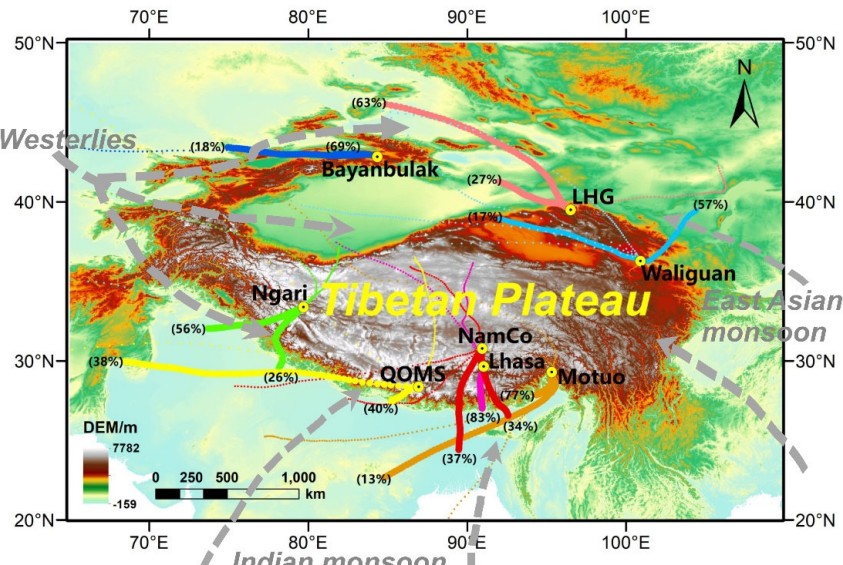

**Figure 10.** The average air mass backward trajectory clusters during the eight field campaigns in the Tibetan Plateau and its surroundings in our study (The geographical base map is created with ArcGIS). The major trajectory clusters belong to each field campaign are displayed using the relatively large solid circles in different colors with contributions marked in the corresponding brackets, while the rest clusters with less contributions are exhibited in small dots.




 **Tables**

**Table 1.** Detailed information about the full name and geographic characteristic of observation
station, sample period, online instruments, and corresponding references during each aerosol field
measurement campaigns over the Tibetan Plateau and its surroundings in this study.

| Station | Full Station Name | Lat. (°N) | Long. (°E) | Alt. (m) | Sample Period | HR-ToF-AMS MS | HR-ToF-AMS PToF | SMPS | PAX | Aethalometer | CCN-100 | References |
|---|---|---|---|---|---|---|---|---|---|---|---|---|
| QOMS | Qomolangma Station for Atmospheric and Environmental Observation and Research, Chinese Academy of Sciences | 28.36 | 86.95 | 4276 | 12 April to 12 May 2016 | √ | √ | √ | √ | √ | | Zhang et al. (2018) An et al. (2019) Xu et al. (2020) Zhang et al. (2021) Xu et al. (2022) |
| Motuo | Motuo County, Linzhi City, Tibet Autonomous Region, China | 29.30 | 95.32 | 1305 | 26 Mar to 22 May 2021 | √ | √ | √ | √ | | √ | This study |
| NamCo | Nam Co Station for Multisphere Observation and Research, Chinese Academy of Sciences | 30.77 | 90.95 | 4730 | 31 May to 1 July 2015 | √ | | | | √ | | Xu et al. (2018) Zhang et al. (2021) |
| Ngari | Ngari Station for Desert Environment Observation and Research, Chinese Academy of Sciences | 33.39 | 79.70 | 4270 | 1 Jun to 5 Jul 2022 | √ | √ | | √ | | | This study |
| Waliguan | China Global Atmospheric Watch Baseline Observatory, Mount Waliguan Base | 36.28 | 100.90 | 3816 | 1 July to 31 July 2017 | √ | √ | | √ | √ | √ | Zhang et al. (2019) Zhang et al. (2020) Xu et al. (2020) Zhang et al. (2021) Xu et al. (2022) |
| LHG | Qilian Observation and Research Station of Cryosphere and Ecologic Environment, Chinese Academy of Sciences | 39.50 | 96.51 | 4180 | 4 August to 29 August 2020 | √ | | √ | | | √ | This study |
| Bayanbulak | Bayanbulak Town, Hejing County, Bayingolin Mongolian Autonomous Prefecture, Xinjiang Uygur Autonomous Region, China | 42.83 | 84.35 | 2454 | 29 August to 26 September 2021 | √ | √ | | | | | This study |
| Lhasa | Lhasa City, Tibet Autonomous Region, China | 29.65 | 91.03 | 3650 | 31 August to 26 September 2019 | √ | √ | √ | √ | | | Zhao et al. (2022) |




**Table 2.** Summary of the average values measured with various instruments during the eight aerosol
field measurement campaigns in the TP and its surroundings in this study.

| Measurement items | QOMS | Motuo | NamCo | Ngari | Waliguan | LHG | Bayanbulak | Lhasa |
|---|---|---|---|---|---|---|---|---|
| **HR-ToF-AMS measurements** | | | | | | | | |
| $PM_1$ mass conc. ($\mu g\ m^{-3}$) | 4.4 | 5.7 | 2.0 | 4.1 | 9.1 | 3.0 | 1.9[a] | 4.7 |
| $PM_1$ chemical compositions (%) | | | | | | | | |
|   OA | 54.4 | 57.0 | 68.0 | 56.3 | 34.5 | 23.1 | 38.4 | 82.6 |
|   Sulfate | 9.3 | 21.1 | 15.0 | 19.8 | 38.1 | 46.0 | 41.6 | 3.4 |
|   Nitrate | 5.1 | 2.3 | 2.0 | 5.7 | 8.1 | 5.7 | 5.4 | 5.4 |
|   Ammonium | 5.8 | 7.3 | 7.0 | 9.2 | 15.2 | 14.4 | 13.6 | 4.7 |
|   Chloride | 0.4 | 0.2 | 0 | 0.4 | 1.1 | 1.2 | 1.0 | 0.7 |
|   BC | 25.0 | 12.1 | 8.0 | 8.6 | 3.0 | 9.7 | N/A | 3.1 |
| Peak diameter in mass size distribution (nm) | | | | | | | | |
|   OA | 510.2 | 430.5 | | 584.4 | 405.5 | | 350.8 | 228.1 |
|   SNA | 510.2 | 471.9 | | 634.5 | 504.7 | | 379.6 | 250.0 |
| OA components (%) | | | | | | | | |
|   MO-OOA | 42.2 | 36.9 | 59.0 | 43.7 | 34.4 | 24.9 | 66.3 | |
|   LO-OOA | | 46.9 | 41.0 | 28.5 | | 75.1 | 33.7 | |
|   OOA | | | | | | | | 34.0 |
|   BBOA | 3.9 | 16.2 | | 27.8 | 18.3 | | | 13.8 |
|   agBBOA | | | | | 40.4 | | | |
|   NOA | 13.9 | | | | | | | |
|   HOA | | | | | 6.9 | | | 16.9 |
|   COA | | | | | | | | 35.3 |
| OA elemental ratios | | | | | | | | |
|   O/C | 1.19 | 0.99 | 1.07 | 0.98 | 0.99 | 1.14 | 0.69 | 0.44 |
|   H/C | 1.29 | 1.55 | 1.48 | 1.33 | 1.41 | 1.05 | 1.52 | 1.76 |
|   OM/OC | 2.70 | 2.48 | 2.57 | 2.44 | 2.45 | 2.62 | 2.09 | 1.74 |
|   N/C | 0.030 | 0.020 | 0.016 | 0.019 | 0.008 | 0.011 | 0.026 | 0.001 |
| **SMPS measurements** | | | | | | | | |
|   Number conc. ($cm^{-3}$) | 709.3 | 1639.2 | | | | 1462.0 | | 3994.4 |
|   Peak diameter in PNSD (nm) | 109.4 | 131.0 | | | | 42.9 | | 28.9 |
| **PAX measurements** | | | | | | | | |
|   $B_{scat}$ ($Mm^{-1}$) | 121.9 | 44.9 | | 8.9 | 36.3 | | | 2.1 |
|   $B_{abs}$ ($Mm^{-1}$) | 10.8 | 7.0 | | 3.6 | 4.1 | | | 1.9 |
|   $B_{ext}$ ($Mm^{-1}$) | 132.7 | 51.9 | | 12.6 | 40.4 | | | 4.0 |
|   SSA | 0.89 | 0.83 | | 0.67 | 0.86 | | | 0.52 |
| **Aethalometer measurements** | | | | | | | | |
|   $B_{abs,370}$ ($Mm^{-1}$) | 13.40 | | 3.25 | | 2.66 | | | |
|   Absorption Ångström exponent | 1.73 | | 1.28 | | 1.12 | | | |
|   $B_{abs,BrC,370}$ ($Mm^{-1}$) | 4.42 | | 0.69 | | 0.60 | | | |
|   $B_{abs,BC,370}$ ($Mm^{-1}$) | 8.94 | | 2.56 | | 2.06 | | | |
|   $fB_{abs,BrC,370}$ (%) | 33.1 | | 21.3 | | 22.4 | | | |
|   $fB_{abs,BC,370}$ (%) | 66.9 | | 78.7 | | 77.6 | | | |
| **CCN-100 measurements ($cm^{-3}$)** | | | | | | | | |
|   CCN number conc. (*SS* 0.2%) | | 974.0 | | | 507.0 | 83.9 | | |
|   CCN number conc. (*SS* 0.4%) | | 1142.6 | | | 805.1 | 344.3 | | |
|   CCN number conc. (*SS* 0.6%) | | 1240.1 | | | 1073.3 | 429.9 | | |
|   CCN number conc. (*SS* 0.8%) | | 1296.5 | | | 1230.6 | 480.8 | | |
|   CCN number conc. (*SS* 1.0%) | | 1337.9 | | | 1336.6 | 516.1 | | |

[a]only non-refractory $PM_1$ is reported at Bayanbulak due to the absence of BC observation.





**Table 3.** Summary of the average $PM_1$ mass concentrations (μg m$^{-3}$) measured by the Aerodyne
AMSs at various high-altitude and remote sites worldwide.

| Observation Sites | Latitude (°N) | Longitude (°E) | Altitude (m a.s.l.) | $PM_1$ mass (μg m$^{-3}$) | References |
|---|---|---|---|---|---|
| QOMS, China | 28.36 | 86.95 | 4276 | 4.4 | This study & Zhang et al. (2018) |
| Motuo, China | 29.30 | 95.32 | 1305 | 5.7 | This study |
| NamCo, China | 30.77 | 90.95 | 4730 | 2.0 | This study & Xu et al. (2018) |
| Ngari, China | 33.39 | 79.70 | 4270 | 4.1 | This study |
| Waliguan, China | 36.28 | 100.90 | 3816 | 9.1 | This study & Zhang et al. (2019) |
| LHG, China | 39.50 | 96.51 | 4180 | 3.0 | This study |
| Bayanbulak, China | 42.83 | 84.35 | 2454 | 1.9[a] | This study |
| Lhasa, China | 29.65 | 91.03 | 3650 | 4.7 | This study & Zhao et al. (2022) |
| NamCo, China | 30.77 | 90.95 | 4730 | 1.06 | Wang et al. (2017) |
| Mt. Yulong, China | 27.20 | 100.20 | 3410 | 5.7 | Zheng et al. (2017) |
| Menyuan, China | 37.61 | 101.26 | 3295 | 11.4 | Du et al. (2015) |
| Mt. Wuzhi, China | 18.84 | 109.49 | 958 | 10.9 | Zhu et al. (2016) |
| Mt. Jungfraujoch, Switzerland | 46.55 | 7.98 | 3580 | 0.55 | Fröhlich et al. (2015) |
| Mt. Jungfraujoch, Switzerland | 46.55 | 7.98 | 3580 | 2.24 | Zhang et al. (2007a) |
| Mt. Bachelor, USA | 43.98 | -121.69 | 2800 | 15.10 | Zhou et al. (2017) |
| Mt. Whistler, Canada | 50.01 | -122.95 | 2182 | 1.91 | Sun et al. (2009) |
| Mt. Cimone, Italy | 44.18 | 10.70 | 2165 | 4.5 | Rinaldi et al. (2015) |
| Puy de Dôme, France | 45.77 | 2.95 | 1465 | 5.58 | Freney et al. (2011) |
| Sub-Antarctic Bird Island | -54.00 | -38.04 | | 0.46 | Schmale et al. (2013) |
| Mace Head, Ireland | 53.30 | -9.80 | | 1.53 | Zhang et al. (2007a) |
| Hyytiala, Finland | 61.90 | 24.30 | | 2.04 | Zhang et al. (2007a) |
| Storm Peak, USA | 40.50 | -106.70 | | 2.11 | Zhang et al. (2007a) |
| Duke Forest, USA | 36.00 | -79.10 | | 2.82 | Zhang et al. (2007a) |
| Chebogue, Canada | 43.80 | -66.10 | | 2.91 | Zhang et al. (2007a) |
| Okinawa Island, Japan | 26.87 | 33.51 | | 7.89 | Jimenez et al. (2009) |
| Fukue Island, Japan | 32.69 | 128.84 | | 12.03 | Takami et al. (2005) |
| Cheju Island, Korea | 33.51 | 126.50 | | 10.66 | Jimenez et al. (2009) |

[a]only non-refractory $PM_1$ is reported at Bayanbulak due to the absence of BC observation.