# Peer review of "High-resolution physicochemical dataset of atmospheric"

_Earth System Science Data, 2023_

## Author Response (AR1)

We appreciate the reviewers for their valuable comments and suggestions for our manuscript. We have revised our manuscript accordingly and our response point-by-point are presented below. Note that the comments are in **black**, and our responses are in **blue**, while the corresponding revise in the manuscript are in **red**.

**Reviewer #1**

The manuscript presents a comprehensive study on atmospheric aerosols in the Tibetan Plateau (TP). The paper details the collection and analysis of aerosol data from multiple sites across TP, emphasizing the physicochemical and optical properties of aerosols. It discusses the impacts of these aerosols on regional climate, ecological balance, and hydrological cycles, highlighting the significance of this research in understanding aerosol behavior in high-altitude areas. The paper is well-structured and presents complex scientific data in an accessible manner. In conclusion, the manuscript is a significant contribution to atmospheric science, particularly in understanding aerosols' role in climatic effect on the Tibetan Plateau. With minor enhancements, it would be a valuable addition to the Journal of Earth Science System Data.

Thank you sincerely for your constructive and positive comments. We have carefully revised our manuscript point-by-point.

General comments:

1. The manuscript's primary concern at present is its language. Engaging a native English speaker to thoroughly refine and enhance the language used throughout the document would be highly beneficial.

   Our manuscript has been revised carefully and the language is polished throughout the manuscript. We hope the language now fits the requirements of the ESSD.

2. The methodology for collecting and analyzing the aerosol data appears robust and thorough. The use of high-resolution instruments and multiple sites enhances the study's reliability. However, it would be beneficial to discuss any limitations or potential biases in the data collection process.

   The measurement uncertainties for each instrument are reported in section 3.1 in the updated manuscript (line 272-277) as follows.

   "The measurement uncertainties for each instrument are difficult to quantify based on data from a single instrument. The uncertainty showing below are referred from other studies: <30% for HR-ToF-AMS (Jimenez et al., 2016) and SMPS, <40% for Aethalometer (Backman et al., 2017), <10% for PAX (Selimovic et al., 2018), and <25% for CCNC (Rose et al., 2008)."

Minor comments:

1. Line 34: "...ecological, environmental security, and the hydrological cycle."

   Revised as the reviewer suggested.

2. Line 48: "...particle light absorption from different carbonaceous substances such as black carbon and brown carbon..."

   Revised as the reviewer suggested.

3. Line 62: "The Tibetan Plateau (TP), with a mean altitude of over 4000 m a.s.l..."

   Revised as the reviewer suggested.

4. Line 75: "As the most complex and important components in the atmosphere, atmospheric aerosols..."

   Revised as the reviewer suggested.

5. Line 110: "...aerosol physical, chemical, and optical properties..."

   Revised as the reviewer suggested.

6. Line 170: "Continuous observations of atmospheric aerosol chemistry have been conducted..."

   Revised as the reviewer suggested.

7. Line 193: "Motuo County, located in the lower reaches of the Yarlung Tsangpo River and the southern..."

   Revised as the reviewer suggested.

8. Line 265: "Online-based observations of atmospheric aerosols were carried out..."

   Revised as the reviewer suggested.

9. Line 866: "...the data reveal seasonal variations in aerosol properties."

   Revised as the reviewer suggested.

10. Line 853: "...which will contribute to our understanding of aerosol-climate interactions."

    Revised as the reviewer suggested.

11. Line 812: "This study represents a stepping stone towards detailed aerosol characterization."

Revised as the reviewer suggested.

12. Line 887: "Acknowledgments: We extend our thanks to the various teams and individuals..."

Revised as the reviewer suggested.

13. Figure 2: Are the concentrations of PM1 at each site converted to standard conditions?

The concentrations in this study are all shown in ambient conditions. This information is supplied in the caption of figure 2.

14. Figure 4: The color of each line is easy to confused with the component of AMS species in the manuscript. It is better to change them to another group of colors. In addition, the diurnal variation of Namco is shift with other lines.

Done. The color for each line has been changed.

[Figure]

15. Figure 6: Although there are distinct contributions of MO-OOA and LO-OOA at different site, these two components at different sites could be different with each other as illustrated with their different O/C ratio. How can these species be compared among them? Is there any suggestion for this kind of comparison?

We agree with that the OOA factors at different sites could be different despite the same name abbreviation. This is due to the limitations of the methodology of PMF. The PMF decomposes the factors of OA based on a few principles including its oxidation level represented by O/C ratio. For primary factors, there are also other makers for identifying its source, however, for secondary factors, the main principle is based on O/C ratio. Therefore, there are typically two secondary OA factors including one less oxidized and one more oxidized OA throughout the HR-AMS community. The difference on O/C of each OOA factor among different sites should be different on their chemical composition and physical properties. However, there are out of the range of this study. For clarifying in our manuscript (line 523-524), the difference properties of each OOA factor among the sites are mentioned as follows.

"Note that the properties of each OOA factor could be different despite the same name across the locations in the TP."

**Reviewer #2**

This manuscript offers a comprehensive examination of the physicochemical and optical properties of atmospheric aerosols over the Tibetan Plateau (TP), employing a broad array of high-resolution, real-time measurements across several key locations. It ambitiously spans a significant and topographically complex area vital for regional and global climate understanding, a region notably under-studied due to logistical and environmental hurdles. The resulting data aims to significantly refine aerosol radiative forcing estimations in climate models for the TP area.

The authors have detailed their methods, from deploying advanced instruments to applying sophisticated data analysis techniques, offering a rich analysis of aerosol dynamics across various TP climates and environments. Moreover, the study explores the broader implications of these findings on climate modeling and potential impacts on the environment.

Despite the manuscript's comprehensive approach and its potential significant impact, areas requiring further clarification or detail could enhance its scientific contribution before it can be accepted.

Thank you sincerely for your constructive and positive comments. We have carefully revised our manuscript point-by-point.

The following are specific comments and suggestions for the authors:

1. The limited coverage of seasonal variability due to short-term observations raises questions about the generalizability of the findings across the TP's complex climate system, influenced by monsoonal patterns. This should be clear in this manuscript.

   Agree. These limitations of short-term observations across the TP in our study have been emphasized in section 5 (Dataset limitations and applications) (line 741-749) as follows. The reasons for these limitations are also summarized afterward.

   "The primary limitation stems from the short and inconsistent measurement periods across different observational years and seasons at different sites, impeding robust comparisons of aerosol properties across the TP. This constraint also hampers the ability to ascertain long-term and seasonal variation characteristics and limits the application of current data and findings to different seasons. The harsh natural environments, challenging weather conditions, limited logistical support and high-resolution instruments, and stringent instrumental requirements (such as the necessity for comprehensive field stations with stable power supply) presented significant challenges during our field observations in these remote TP regions. It is worth noting that online HR-ToF-AMS observations, such as the one we conducted, are predominantly short-term and intensive observations carried out worldwide due to the stability issues and its challenging maintenance required for long-term measurements."

2. Mention of the Positive Matrix Factorization (PMF) and HYSPLIT models lacks a detailed rationale for their selection and configuration. An elaboration on the decision-making process and any comparative analysis with alternatives would be beneficial.

We acknowledge that providing a detailed explanation of the decision-making processes of PMF and HYSPLIT in the manuscript would be highly beneficial. However, due to constraints on manuscript length and the primary objective of presenting the dataset in this paper, we are unable to delve into these specifics extensively. Notably, such information was previously detailed in our publication for three specific sites (NamCo, QOMS, and Waliguan). To address this in the revised manuscript, we have indicated that this information is available in our prior publication (line 329-332).

"The details of the PMF solution determination for each site are not presented here but can be referenced in our previous publication for select campaigns (Xu et al., 2018; Zhang et al., 2018; 2019)."

The manuscript already includes details on the parameters set for the HYSPLIT analysis, such as the sampling site's height and the frequency of air mass backward trajectories. An additional sentence is incorporated in the updated manuscript to elucidate the process of determining cluster results (line 704-705).

"The cluster analysis on the trajectories was based on the total spatial minimum variance method."

3. The manuscript touches on regional variations but falls short of quantitatively attributing aerosol concentrations to specific sources. Could additional data or analytical methods improve direct source apportionment?

Thank you for your valuable suggestion. However, attributing aerosols to specific sources based on our current observations is challenging. The results presented in the manuscript qualitatively indicate the direction of aerosol transport and the chemical composition of transported aerosols at the sampling site, derived from a combination of trajectory analysis and chemical composition measurements. For a more in-depth source apportionment analysis, a model study in conjunction with satellite observations would be required, which falls beyond the scope of our current study. Such detailed investigations will be pursued in future studies, potentially focusing on a specific site of our study to provide more comprehensive insights.

4. The research's potential to enhance climate model predictions is intriguing. A more detailed discussion on integrating these findings into existing climate models, specifically which model parameters or processes this dataset could refine, would be valuable.

Agree. One sentence is provided in the updated manuscript to emphasize this information (line 772-775) as follows.

"Furthermore, these observations are invaluable for model simulation and verification, providing a wealth of data points that can be utilized for assessing aerosol loading, chemical composition, size distribution, and other parameters essential for model accuracy and validation."

5. The inherent limitations and uncertainties of observational studies, especially in challenging environments like the TP, warrant a more detailed discussion. The manuscript would benefit from addressing the spatial and temporal data coverage limitations and their potential impact on the study's conclusions.

The limitations and uncertainties of our studies across the TP outlined in section 5 have been further addressed in the updated manuscript (line 760-763). Given the scarcity of observatories in the TP, the current observations conducted across multiple locations represent the most comprehensive effort possible. We have maximized our research efforts within the available observatories in the TP. The site selection for each observatory actually followed the requirement for representing the specific geographic and climatic features unique in the TP.

"Furthermore, assessing the representativeness of each observation for the spatial scale is particularly challenging due to the limited number of observatories across the TP. Actually, these observatories have been strategically established based on the representation of specific geographic and climatic features."

6. Finally, the entire text requires further revision for English expression by a native English speaker.

Our manuscript has been revised carefully and the language is polished throughout the manuscript. We hope the language now fits the requirements of the ESSD.

---

## Author Response (AR2)

Dear Prof. David Carlson and Polina Shvedko,

Thank you very much for your interest on our work. We have revised our manuscript following your comments and suggestions. Point-by-point responses are shown below. Hope the updated manuscript fits the requirements of ESSD.

Best regards,

Jianzhong Xu

A good product with value enhanced by exposure through data sharing. Thank you for using ESSD. Additional private note (visible to authors and reviewers only):
Please work closely with typesetters and proof-readers on remaining language and technical uncertainties. Figure 10, for example, remains difficult to read and interpret. Perhaps clarify intent: not a perfect data set for valid reasons but sharing enhances quality and exposes both value and difficulties. Overall: a good product, thank you for using ESSD.

We will closely work with typesetters and proof-readers on polishing our language and technical uncertainties. For figure 10, the color and caption are revised correspondingly. The mention on the mean of sharing on dataset is shown below (line 778-779 in manuscript).

"Although the data set is not perfect for valid reasons, sharing it improves quality and exposes both value and difficulties."

Please ensure that the colour schemes used in your maps and charts allow readers with colour vision deficiencies to correctly interpret your findings. Please check your figures using the Coblis – Color Blindness Simulator (https://www.color-blindness.com/coblis-color-blindness-simulator/) and revise the colour schemes accordingly.

The figures are corrected in Coblis.